# Evolution of host-microbe cell adherence by receptor domain shuffling

EmilyClare P Baker[1], Ryan Sayegh[1], Kristin M Kohler[1], Wyatt Borman[1], Claire K Goodfellow[1,2], Eden R Brush[1], Matthew F Barber[1,2]*

[1]Institute of Ecology and Evolution, University of Oregon, Eugene, United States; [2]Department of Biology, University of Oregon, Eugene, United States

**Abstract** Stable adherence to epithelial surfaces is required for colonization by diverse host-associated microbes. Successful attachment of pathogenic microbes to host cells via adhesin molecules is also the first step in many devastating infections. Despite the primacy of epithelial adherence in establishing host-microbe associations, the evolutionary processes that shape this crucial interface remain enigmatic. Carcinoembryonic antigen-related cell adhesion molecules (CEACAMs) encompass a multifunctional family of vertebrate cell surface proteins which are recurrent targets of bacterial adhesins at epithelial barriers. Here, we show that multiple members of the primate CEACAM family exhibit evidence of repeated natural selection at protein surfaces targeted by bacteria, consistent with pathogen-driven evolution. Divergence of CEACAM proteins between even closely related great apes is sufficient to control molecular interactions with a range of bacterial adhesins. Phylogenetic analyses further reveal that repeated gene conversion of CEACAM extracellular domains during primate divergence plays a key role in limiting bacterial adhesin host tropism. Moreover, we demonstrate that gene conversion has continued to shape CEACAM diversity within human populations, with abundant human CEACAM1 variants mediating evasion of adhesins from pathogenic *Neisseria*. Together this work reveals a mechanism by which gene conversion shapes first contact between microbes and animal hosts.

*For correspondence:
mfbarber@uoregon.edu

**Competing interest:** The authors declare that no competing interests exist.

## Editor's evaluation

In this very interesting manuscript, Baker et al., investigates the molecular evolution in primates of one protein family, the CEACAMs, that are a recurrent target of bacterial surface adhesions at epithelial surfaces. They show that multiple members of this gene family have experienced repeated episodes of positive selection in primates, especially in the N-terminal domains that are associated with protein binding and go on to evaluate the functional consequences of these evolutionary changes. These findings are important to inform our understanding of the co-evolution of interactions between microbes and their mammalian hosts.

## Introduction

Epithelial surfaces are typically the initial point of contact between metazoans and microbes (*Brown and Clarke, 2017*). As such, host factors at this barrier play an important role in facilitating or deterring microbial colonization. Bacterial attachment to epithelial surfaces is often mediated by a broad class of surface proteins termed adhesins (*Kline et al., 2009*). In addition to permitting the growth and colonization of commensal microbes, adhesins are also key virulence factors for many pathogenic bacteria. Adhesin-mediated adherence to host cells is often required for other downstream processes including biofilm formation, epithelial invasion, and the delivery of toxic effectors into host cells (*Kline et al., 2009*; *Sadarangani et al., 2011*; *Figure 1*). Microbial adherence can also trigger epithelial

**eLife digest** Trillions of bacteria live in and on the human body. Most of them are harmless but some can cause serious infections. To grow in or on the body, bacteria often attach to proteins on the surface of cells that make up the lining of tissues like the gut or the throat. In some cases, bacteria use these proteins to invade the cells causing an infection. Genetic mutations in the genes encoding these proteins that protect against infection are more likely to be passed on to future generations. This may lead to rapid spread of these beneficial genes in a population.

A family of proteins called CEACAMs are frequent targets of infection-causing bacteria. These proteins have been shown to play a role in cancer progression. But they also play many helpful roles in the body, including helping transmit messages between cells, aiding cell growth, and helping the immune system recognize pathogens. Scientists are not sure if these multi-tasking CEACAM proteins can evolve to evade bacteria without affecting their other roles.

Baker et al. show that CEACAM proteins targeted by bacteria have undergone rapid evolution in primates. In the experiments, human genes encoding CEACAMs were compared with equivalent genes from 19 different primates. Baker et al. found the changes in human and primate CEACAMs often occur through a process called gene conversion. Gene conversion occurs when DNA sections are copied and pasted from one gene to another. Using laboratory experiments, they showed that some of these changes enabled CEACAM proteins to prevent certain harmful bacteria from binding.

The experiments suggest that some versions of CEACAM genes may protect humans or other primates against bacterial infections. Studies in natural populations are needed to test if this is the case. Learning more about how CEACAM proteins evolve and what they do may help scientists better understand the role they play in cancer and help improve cancer care. Studying CEACAM evolution may also help scientists understand how bacteria and other pathogens drive protein evolution in the body.

cell signaling cascades, further shaping host responses to resident and invasive microbes. Despite the fundamental importance of epithelial adherence for bacterial colonization and infectious disease pathogenesis, the dynamics of these interactions between host surface proteins and bacterial adhesions over evolutionary timescales remain a mystery. Theory predicts that exploitation of host proteins by pathogens places a significant burden on host populations, driving selection for beneficial mutations that limit microbial invasion or virulence. From a microbial perspective, host defenses can also pose an existential threat resulting in reciprocal adaptation to enhance colonization, growth, and transmission (*Aleru and Barber, 2020*; *Brockhurst et al., 2014*; *Hamilton et al., 1990*; *Van Valen, 1973*). However, pathogens hijack many host factors not directly involved in immunity, possibly limiting their adaptive potential in response to pathogen interaction. For example, epithelial surface proteins are not only essential for interacting with the environment but also serve crucial cellular and physiological functions including barrier maintenance, cell-cell communication, as well as coordinating host physiological and developmental pathways (*Kuespert et al., 2006*). It therefore remains unclear the extent to which such proteins are able to adapt in the face of pathogen antagonism.

Bacterial adhesins interact with a wide range of molecules present on host epithelial surfaces (*Chatterjee et al., 2021*). One important target of bacterial adhesins on vertebrate epithelia are the carcinoembryonic antigen-related cell adhesion molecule (CEACAM) family of proteins (*Gray-Owen and Blumberg, 2006*). There have been multiple independent expansions of this gene family across mammals (*Kammerer and Zimmermann, 2010*; *Pavlopoulou and Scorilas, 2014*) and the human genome encodes 12 functional CEACAM genes as well as several pseudogenes (*Gray-Owen and Blumberg, 2006*; *Kammerer and Zimmermann, 2010*). Collectively, CEACAMs are expressed on nearly all vertebrate epithelial surfaces including the microbe-rich surfaces of the urogenital, respiratory, and gastrointestinal tracts (*Tchoupa et al., 2014*). Epithelial CEACAMs play a variety of roles in cell adhesion as well as intra- and intercellular signaling (*Gray-Owen and Blumberg, 2006*; *Kuespert et al., 2006*; *Tchoupa et al., 2014*). A subset of CEACAMs are also expressed on other cell types, including T-cells and neutrophils where they play important roles in immune signaling and pathogen recognition. CEACAMs typically consist of an extracellular N-terminal IgV-like domain (also termed the N-domain), a variable number of IgC-like domains, and either a GPI anchor or a transmembrane and

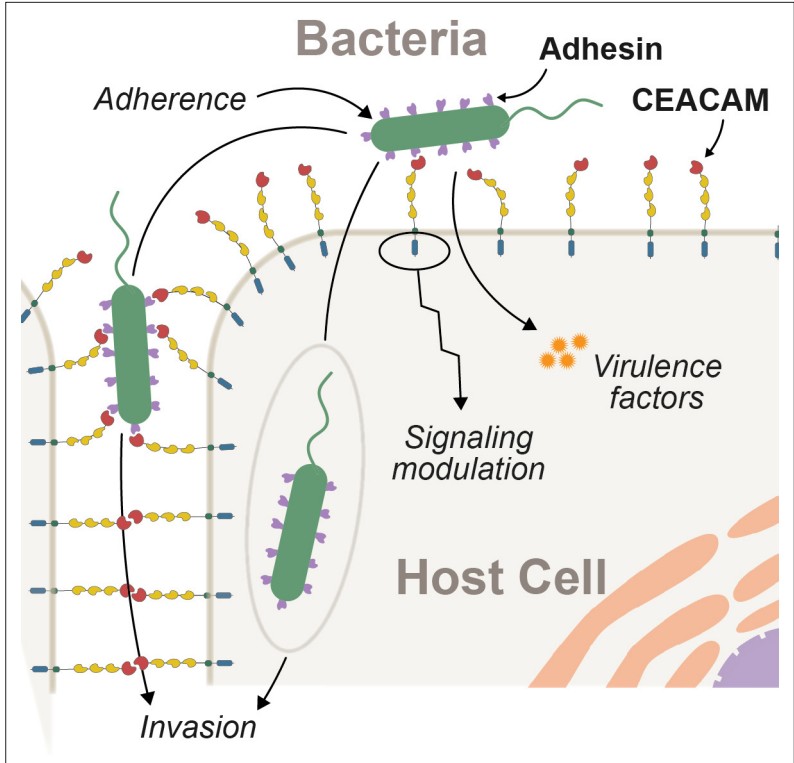

**Figure 1.** Interactions between epithelial carcinoembryonic antigen-associated cell adhesion molecules (CEACAMs) and bacterial adhesins. Bacterial attachment to host cells via adhesin proteins (purple) facilitates epithelial adherence. Adhesins also contribute to pathogenicity by promoting invasion, modulation of host cell signaling pathways, and by promoting the delivery of virulence factors into the host cell cytoplasm.

cytoplasmic domains. The characteristics of this cytoplasmic domain in turn influences the functional properties of CEACAM proteins. Extracellular protein-protein interactions involving CEACAMs have been shown to primarily occur through the extracellular N-domain (*Kuespert et al., 2007*; *Markel et al., 2004*).

While the functions of many CEACAM proteins remain obscure, mammalian CEACAM1, CEACAM5 (also known as CEA), and CEACAM6 have been shown to contribute to immunoregulation, cell-cycle progression, and development (*Gray-Owen and Blumberg, 2006*; *Kuespert et al., 2006*; *Tchoupa et al., 2014*). These CEACAMs, along with CEACAM3, are also notable as recurrent targets of bacterial adhesins (*Gray-Owen and Blumberg, 2006*; *Tchoupa et al., 2014*). A growing number of bacterial genera have been found to target these CEACAM proteins to promote epithelial adherence and host colonization, including *Neisseria*, *Haemophilus*, *Escherichia*, *Fusobacterium*, *Streptococcus*, and *Helicobacter* (*Brewer et al., 2019*; *Gray-Owen and Blumberg, 2006*; *Javaheri et al., 2016*; *Königer et al., 2016*; *van Sorge et al., 2021*). Additionally, several viruses have been reported to bind CEACAMs, including human cytomegalovirus, influenza A, murine coronavirus, and Middle East respiratory syndrome coronavirus (*Chan et al., 2016*; *Hemmila et al., 2004*; *Macmaniman et al., 2014*; *Rahman et al., 2021*). While many of the bacteria reported to target CEACAMs are able to colonize the host as benign commensals, they are also capable of causing serious infection especially in young children. Such infections may prove severely detrimental to host fitness even when not fatal.

Among bacteria, a number of structurally unrelated CEACAM-binding adhesin proteins have been described. This diversity of structures suggests that CEACAM recognition has arisen independently multiple times during bacterial evolution. These structurally diverse proteins include the integral outer membrane Opa proteins in *Neisseria* (*Fox et al., 2014*), the immunoglobulin-type β protein from group B *Streptococcus* (*van Sorge et al., 2021*), as well as other trimeric and globular protein domains (*Bonsor et al., 2018*; *Conners et al., 2008*).

Bacterial CEACAM recognition can lead to several distinct outcomes (*Figure 1*). First, adherence to epithelial CEACAMs can provide a stable habitat to support bacterial growth and proliferation.

In mice, for example, expression of human CEACAM1 is sufficient to establish stable colonization by otherwise human-restricted strains of *Neisseria meningitidis* (*Johswich et al., 2013*). Second, CEACAM binding may facilitate bacterial dissemination through the host epithelium (*Wang et al., 1998*). Third, in the case of the bacterium *Helicobacter pylori*, CEACAM-adhesin interactions promote the translocation of virulence factors into host cells via the type 4 secretion system leading to severe gastritis and stomach ulcers in humans (*Javaheri et al., 2016*; *Königer et al., 2016*). Finally, bacterial adhesins can potentiate CEACAM-mediated signaling cascades to manipulate cellular functions, including preventing immune cell activation (*Gur et al., 2019a*; *Gur et al., 2019b*; *Sadarangani et al., 2011*), increasing cellular adhesion to prevent shedding of infected cells (*Muenzner et al., 2016*; *Muenzner et al., 2010*), and activation of apoptosis (*N'Guessan et al., 2007*). Interactions between adhesins and CEACAMs, particularly bacterial immunoglobulins that appear to mimic CEACAMs, are predicted to further disrupt endogenous CEACAM adhesion and signaling functions (*Macmaniman et al., 2014*; *Moonens, 2018*; *van Sorge et al., 2021*).

Previous work has reported multiple instances of gene gain and loss as well as high levels of sequence divergence among a subset of CEACAM genes (*Adrian et al., 2019*; *Rhesus Macaque Genome Sequencing and Analysis Consortium et al., 2007*; *Kammerer and Zimmermann, 2010*; *Pavlopoulou and Scorilas, 2014*). These findings, coupled with the observation that many CEACAM-binding bacteria possess a narrow host range, suggest that host genetic variation may be a major determinant of bacterial colonization. In the case of CEACAM3, which is expressed exclusively in neutrophils and aids in destruction of CEACAM-binding bacteria, there is compelling evidence that residues at the interface of adhesin binding are evolving rapidly in a manner consistent with positive selection (*Adrian et al., 2019*). Unlike many other characterized mammalian CEACAMs, CEACAM3 appears to be a dedicated immune protein acting as a decoy receptor for CEACAM targeting bacteria (*Bonsignore et al., 2019*). The evolution of epithelial CEACAMs not dedicated to immune defense and the consequences of their evolution for microbial interactions remains unclear.

## Results

### The CEACAM gene family has undergone repeated episodes of positive selection in primates

To assess patterns of primate CEACAM gene evolution, we compiled sequences of human CEACAM orthologs present in publicly available genome databases (*Figure 2—source data 1A-C*). Nineteen representative species were analyzed including four New World monkeys, ten Old World monkeys, and five hominid species. Some orthologs of human CEACAMs were not identified in a subset of primate genomes, likely due to losses or gains of specific CEACAMs along different lineages or incomplete genome assembly. With the exception of CEACAM3, for which additional exons annotated in Old World monkeys were included (detailed in Materials and methods), only genomic sequences that aligned to annotated human exons were used for subsequent phylogenetic analyses. To determine if primate CEACAMs have been subject to positive selection, protein-coding sequences were analyzed using the PAML NS sites package (*Yang, 2007*). This program uses a maximum likelihood framework to estimate the rate of evolution of each gene or codon, expressed as the ratio of normalized nonsynonymous (dN) to synonymous (dS) nucleotide substitutions (dN/dS or $\omega$), under different models of evolution. An excess of nonsynonymous substitutions relative to synonymous substitutions between orthologs can suggest that beneficial mutations have been repeatedly fixed by positive selection. A comparison of models that allow and disallow sites evolving under positive selection ($\omega > 1$) can determine the likelihood that a particular protein-coding sequence has been evolving under positive selection. We found that eight of the twelve primate CEACAM paralogs in our dataset possess genetic signals of positive selection (p-value $\leq 0.05$; *Figure 2—source data 1D*) including CEACAM1, CEACAM3, CEACAM5, and CEACAM6 which have previously been shown to interact with bacterial adhesins (*Gray-Owen and Blumberg, 2006*). In addition, we also identified elevated $\omega$ values for CEACAM7, CEACAM8, CEACAM18, and CEACAM20.

To identify specific amino acid positions that contribute to signatures of positive selection, we analyzed CEACAM sequences using the Bayes empirical Bayes analysis as implemented in the PAML NS sites package, as well as the programs FUBAR and MEME from the HyPhy software package (*Figure 2—source data 1E*). To control for the potential impact of recombination on these inferences,

we used the program GARD to identify potential breakpoints in our datasets and build phylogenies of gene segments based on predicted breakpoints. We then performed phylogenetic analyses using these GARD-informed phylogenies. Our analyses collectively revealed that sites with elevated $\omega$ were concentrated in the N-domain of many CEACAM proteins (*Figure 2A*; *Figure 2—figure supplement 1*). Sites under positive selection in CEACAM18 and CEACAM20 were more dispersed throughout the protein, not localizing to a specific domain. The statistical support for positive selection of CEACAM18 and CEACAM20 in primates was also modest compared to that for other CEACAM proteins.

We next sought to determine the functional impact of divergence at rapidly evolving sites in the CEACAM N-domain. Residues that contribute to protein-protein interactions have been extensively annotated for CEACAM1, involving both host factors and bacterial adhesins. Overlaying sites under positive selection with known adhesin and host protein-binding sites (*Figure 2—source data 1F*) revealed extensive overlap between all three categories (*Figure 2B*) and demonstrates that sites with elevated $\omega$ tend to cluster on the protein-binding surface. Mapping rapidly evolving CEACAM1 residues onto a co-crystal structure of human CEACAM1 bound to the HopQ adhesin from *H. pylori* strain G27 (*Moonens, 2018*), we confirmed that multiple sites fall along the binding interface of the two proteins (*Figure 2C*). In summary, these results demonstrate that multiple primate CEACAM orthologs exhibit signatures of repeated positive selection within the N-domain which facilitates bacterial and host protein interactions.

## CEACAM divergence impairs recognition by multiple bacterial adhesins

To assess how rapid divergence of primate CEACAMs influences recognition by bacterial adhesins, we focused on CEACAM1 which is widely expressed across different cell types (*Gray-Owen and Blumberg, 2006*) and has numerous well-documented microbial interactions (*Figure 2—source data 1F*). Recombinant GFP-tagged CEACAM1 N-domain proteins from a panel of primate species were expressed and purified from mammalian cells (see Materials and methods). Previous studies have demonstrated that the CEACAM N-domain is both necessary and sufficient to mediate interactions with bacterial adhesins (*Javaheri et al., 2016*; *Kuespert et al., 2007*; *Markel et al., 2004*). We focused our experiments on CEACAM1 binding to two distinct classes of bacterial adhesins: HopQ encoded by *H. pylori* and the Opa family adhesins expressed by *Neisseria* species.

The HopQ adhesin is a *H. pylori*-specific outer membrane protein that appears to be universally encoded by *H. pylori* strains and whose interaction with human CEACAM1 has been well characterized (*Bonsor et al., 2018*; *Javaheri et al., 2016*; *Königer et al., 2016*; *Moonens, 2018*). For our assays we used the common *H. pylori* laboratory strains G27 (*Baltrus et al., 2009*), J99 (*Alm et al., 1999*), and Tx30a (ATCC 51932), which have previously been confirmed to bind human CEACAM1 (*Javaheri et al., 2016*). The HopQ proteins encoded by these strains encompass the two major divisions of HopQ diversity, termed Type I and Type II (*Cao and Cover, 2002*; *Javaheri et al., 2016*). Strains G27 and J99 both encode a single copy of a Type I HopQ adhesin, while Tx30a encodes a Type II HopQ adhesin. All strains include extensive divergence in the CEACAM1-binding region (*Bonsor et al., 2018*; *Moonens, 2018*).

Opa proteins are a highly diverse class of adhesins encoded by *Neisseria* species that are structurally distinct from the HopQ adhesin (*Bonsor et al., 2018*; *Fox et al., 2014*; *Moonens, 2018*; *Sadarangani et al., 2011*). For our study we tested CEACAM1 binding to Opa52 and Opa74, which despite their limited sequence identity are both known to bind human CEACAM1. While Opa74 is only known to recognize human CEACAM1, Opa52 also binds CEACAM3, CEACAM5, and CEACAM6 (*Roth et al., 2013*). Because *Neisseria* species typically encode multiple unique phase-variable Opa variants, individual Opa genes from *Neisseria gonorrhoeae* were cloned and expressed heterologously in K12 *Escherichia coli*, which does not bind to CEACAM proteins.

To assess pairwise interactions between primate CEACAMs and bacterial adhesins, we incubated recombinant CEACAM1 N-domain proteins with individual bacterial strains. Bacterial cells were washed, pelleted, and the presence of bound CEACAM1 protein was assessed by western blot. We observe that all bacterial strains tested bind to the human CEACAM1 N-domain, consistent with previous studies (*Figure 3A*). Incubation of *H. pylori* strain G27 with GFP alone fails to yield detectable signal, confirming that binding is CEACAM-dependent (*Figure 3B*). Furthermore, a Δ*hopq* mutant of strain G27 does not exhibit significant CEACAM1 binding, consistent with previous reports that HopQ is the sole CEACAM-binding adhesin present in these strains (*Figure 3—figure supplement 1*).

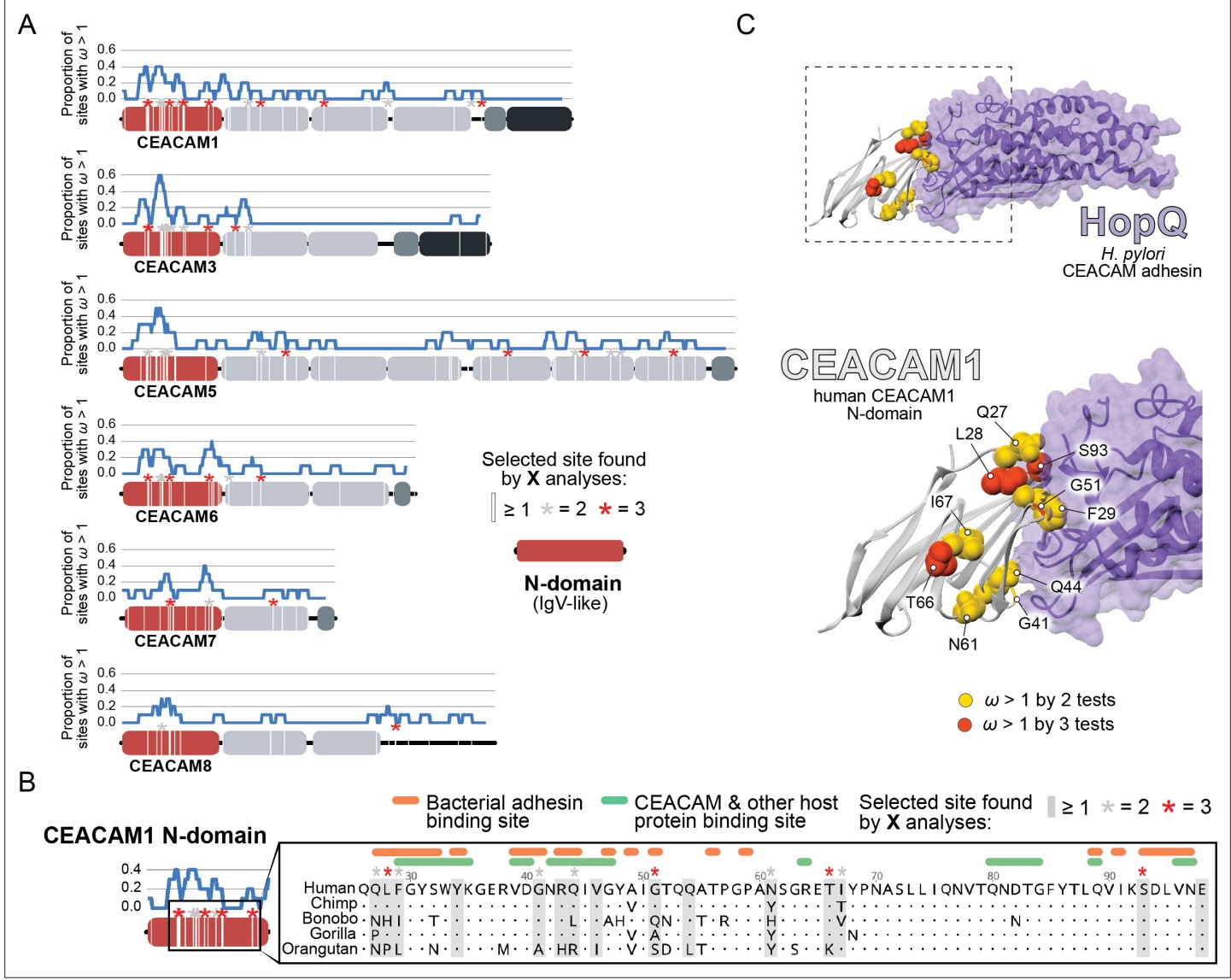

**Figure 2.** Rapid evolution of primate carcinoembryonic antigen-related cell adhesion molecule (CEACAM) N-domains. (**A**) Sites in CEACAM proteins exhibiting elevated $\omega$. Domain structure of CEACAMs outlined in red (N-domain), light gray (IgC-like domains), dark gray (transmembrane domain), and black (cytoplasmic domain). All rapidly evolving sites identified by at least one phylogenetic analysis (PAML, FUBAR, or MEME) are marked by a white line, sites identified by two or three tests signified by gray and red asterisks, respectively. Blue line shows the proportion of rapidly evolving sites identified across a 10 amino acid sliding window. (**B**) Multiple sequence alignment of hominid CEACAM1 residues 26–98. Sites identified as evolving under positive selection and sites known to influence adhesin and host protein binding are highlighted (*Figure 2—source data 1F*). (**C**) Protein co-crystal structure of human CEACAM1 (gray) and the HopQ adhesin (purple) from *Helicobacter pylori* strain G27 (PDB ID: 6GBG). CEACAM1 sites identified as evolving under positive selection by two or more tests are highlighted.

The online version of this article includes the following source data, source code, and figure supplement(s) for figure 2:

**Source code 1.** Code to generate graphs and images for *Figure 2A*.

**Source data 1.** (a) Summary of primate carcinoembryonic antigen-related cell adhesion molecule (CEACAM) sequences used in analyses.

**Source data 2.** Trimmed carcinoembryonic antigen-related cell adhesion molecule (CEACAM) sequences and primate species trees used for evolutionary analyses.

**Source data 3.** Results files for evolutionary analyses.

**Figure supplement 1.** Primate carcinoembryonic antigen-related cell adhesion molecule (CEACAM) evolutionary analysis summary.

**Figure supplement 1—source code 1.** Code to generate graphs and images for *Figure 2—figure supplement 1*.

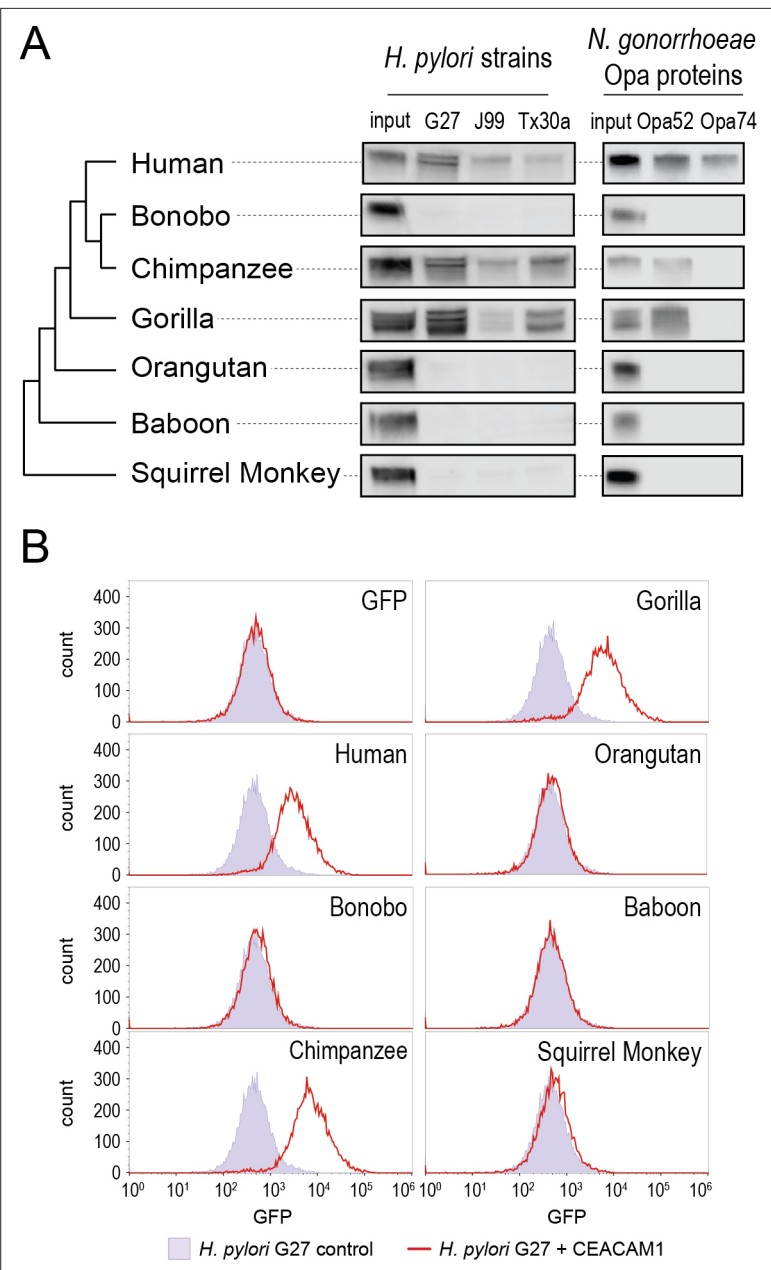

**Figure 3.** Carcinoembryonic antigen-related cell adhesion molecule 1 (CEACAM1) divergence in great apes restricts bacterial adhesin recognition. (**A**) Binding between primate GFP-tagged CEACAM1 N-domain orthologs and bacteria determined by pulldown assays and visualized by western blotting. Input is 10% CEACAM1 protein used in bacterial pulldowns. Primate species relationships indicated by phylogenetic tree. (**B**) Pulldown experiments of *Helicobacter pylori* strain G27 incubated with CEACAM1 N-domain constructs or GFP assayed by flow cytometry. Binding indicated by relative GFP fluorescence. Representative western blot and flow cytometry experiments are depicted. For flow cytometry all tests shown were performed as part of a single experiment using *H. pylori* strain G27 alone as a negative control.

The online version of this article includes the following source data and figure supplement(s) for figure 3:

**Source data 1.** Raw and labeled western blot images for *Figure 3A* and flow cytometry data for *Figure 3B*.

**Figure supplement 1.** *Helicobacter pylori* G27 *Δhopq* pulldown.

**Figure supplement 1—source data 1.** Raw and labeled western blot images for *Figure 3—figure supplement 1*.

Examining non-human CEACAM1 bacterial binding, the chimpanzee CEACAM1 N-domain, which differs from the human protein at four amino acid positions, binds to all adhesin-expressing strains except Opa74. Gorilla CEACAM1, which differs from the human N-domain at five sites (three non-overlapping with chimpanzee), is also unable to bind Opa74 but does bind *H. pylori* strains and Opa52. Orangutan CEACAM1 is unable to interact with any bacterial strain tested, nor do baboon and squirrel monkey. We noted that despite the limited species divergence between bonobos and chimpanzees, bonobo CEACAM1 does not bind any of the tested bacterial strains (*Figure 3A*). Previous studies have found the results of CEACAM-binding assays to be consistent between western blotting and by flow cytometry (*Adrian et al., 2019*; *Javaheri et al., 2016*; *Königer et al., 2016*; *Kuespert et al., 2007*). We confirmed this for our system with *H. pylori* strain G27, using flow cytometry to detect specific binding of GFP-tagged CEACAMs on the bacterial cell surface (*Figure 3B*). These results demonstrate that CEACAM1 N-domain divergence between closely related primate species, even within the great apes, determines bacterial recognition in an adhesin-specific manner.

## Recurrent gene conversion of primate CEACAM N-domains

The inability of *H. pylori* strains or *N. gonorrhoeae* adhesins to bind bonobo CEACAM1 was surprising given bonobo's close phylogenetic relationship to both humans and chimpanzees. While archaic humans are believed to have diverged from our primate relatives at least 5 million years ago, the major divergence between chimpanzees and bonobos occurred only 1–2 million years ago (*Prado-Martinez et al., 2013*). Closer inspection revealed that the bonobo CEACAM1 N-domain sequence is unusually divergent from that of both humans and chimpanzees, while other regions of the coding sequence show higher degrees of identity (*Figure 4—figure supplement 1A*). To investigate bonobo CEACAM1 evolution further, we first validated the bonobo CEACAM1 N-domain sequence present in our bonobo reference genome through comparison of assemblies and sequencing reads from multiple bonobo individuals as well as through direct Sanger sequencing of the CEACAM1 N-domain from bonobo genomic DNA (see Materials and methods). Having confirmed the identity of the bonobo CEACAM1 reference sequence, we compared this gene to sequences from other hominids. Relative to its orthologs in humans and chimpanzees, bonobo CEACAM1 differs at nearly 20% of sites in the N-domain whereas humans and chimpanzees differ at only about 4% of sites. In contrast, outside of the N-domain bonobo CEACAM1 diverges from humans and chimpanzees at approximately 2% of sites, while human and chimpanzee CEACAM1 differ at around 1% of sites. We also noted that the number of divergent sites between bonobo and human in the N-domain (18 residues) is nearly identical to the number of divergent sites between bonobo and chimpanzee (20 residues), despite the closer phylogenetic relationship between bonobos and chimpanzees. In fact, the divergence between the bonobo and chimpanzee CEACAM1 N-domains is greater than that between chimpanzee and the earliest diverging member of the hominid clade, orangutan (81% versus 83% amino acid identity, respectively). A comparison of N-domain sequences for CEACAM5, another rapidly evolving CEACAM, further highlights the extreme divergence of bonobo CEACAM1. Between human CEACAM5 and the bonobo and chimpanzee CEACAM5 sequences, there are only ten and nine amino acid changes respectively, while bonobo and chimpanzee differ at only five sites along the entire length of the N-domain (*Figure 4—figure supplement 1B*).

The degree of divergence within the N-domain of bonobo CEACAM1 suggests processes other than sequential accumulation of single nucleotide mutations could be responsible. One mechanism by which this could occur is through gene conversion, a form of homologous recombination in which genetic material from one location replaces sequence in a non-homologous location, often with substantial sequence similarity (*Chen et al., 2007*). Gene conversion can provide an important source of genetic novelty and a mechanism that can accelerate adaptation (*Bittihn and Tsimring, 2017*; *Daugherty and Zanders, 2019*; *Gendreau et al., 2021*). To determine if inter-locus recombination has shaped the evolution of CEACAM genes in primates, we looked for evidence of discordance between species and gene trees. Gene-species tree discordance can be an indication of multiple evolutionary processes, including a history of gene conversion between paralogs. In a maximum likelihood-based phylogeny of full-length CEACAM-coding sequences, clades containing single CEACAM paralogs were inferred with robust statistical support (*Figure 4A*, *Figure 4—figure supplement 2*). In general, the relationships between CEACAM homologs are inferred with high confidence and reflected species relationships as expected for the divergence of orthologous-coding sequences.

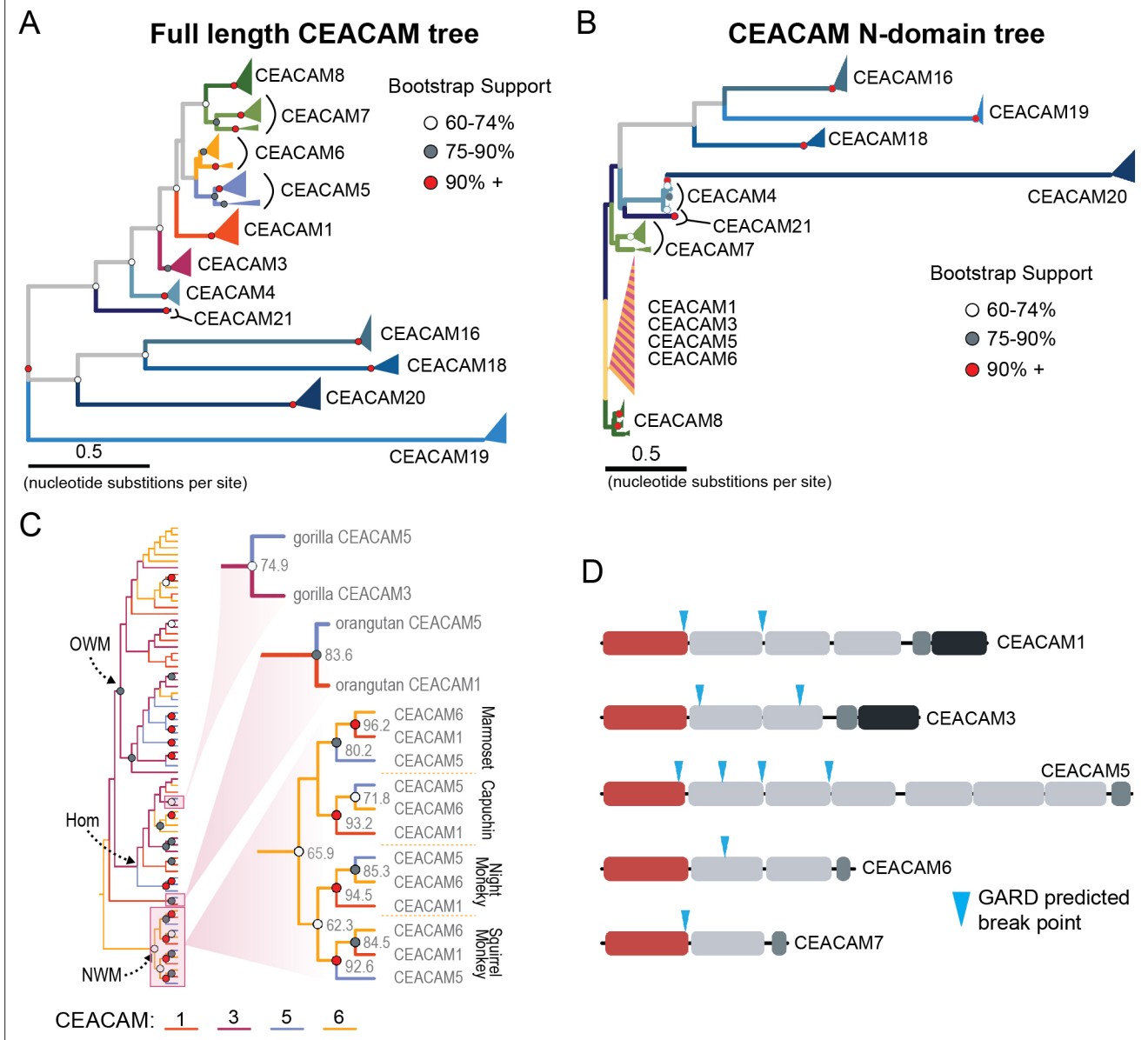

**Figure 4.** Recurrent episodes of gene conversion among adhesin-binding carcinoembryonic antigen-related cell adhesion molecules (CEACAMs). (**A**) Maximum likelihood-based phylogeny of full-length primate CEACAM protein-coding sequences. (**B**) Phylogeny of the IgV-like (N-domain) of primate CEACAM proteins. (**C**) Expanded cladogram view of the clade containing the N-domains of CEACAM1, CEACAM3, CEACAM5, and CEACAM6 from panel B. Arrows indicate nodes designating clades for Old World monkeys (OWM), hominoids (Hom), and New World monkeys (NWM). Specific subclades, gorilla CEACAM3 and CEACAM5, orangutan CEACAM5 and CEACAM1, and NWM are further magnified and highlighted with bootstrap support at nodes. (**D**) Domain structures of CEACAM proteins predicted to have undergone recombination by GARD analysis with sites of predicted breakpoints highlighted (blue arrows). CEACAM N-domains are denoted in red.

The online version of this article includes the following source data, source code, and figure supplement(s) for figure 4:

**Source code 1.** Code to generate images for *Figure 4D*.

**Source data 1.** Sequence alignments of trimmed carcinoembryonic antigen-related cell adhesion molecule (CEACAM) sequences used for phylogenetic reconstructions.

**Figure supplement 1.** Alignment of human-pan carcinoembryonic antigen-related cell adhesion molecule (CEACAM) sequences.

**Figure supplement 2.** Expanded full-length carcinoembryonic antigen-related cell adhesion molecule (CEACAM) tree.

**Figure supplement 3.** Expanded carcinoembryonic antigen-related cell adhesion molecule (CEACAM) N-domain tree.

**Figure supplement 4.** Expanded view of carcinoembryonic antigen-related cell adhesion molecule (CEACAM)1,3,5,6 N-domain clade.

*Figure 4 continued on next page*

*Figure 4 continued*

**Figure supplement 5.** Expanded carcinoembryonic antigen-related cell adhesion molecule (CEACAM) IgC domains tree.

**Figure supplement 6.** Expanded carcinoembryonic antigen-related cell adhesion molecule (CEACAM) cytoplasmic domain tree.

To determine if there have been domain-specific instances of gene conversion, we constructed phylogenetic trees of individual CEACAM domains. Typically, we expect paralogs to form clearly defined clades reflecting species divergence. This is the pattern we observe for full-length CEACAM-coding sequences, indicating that overall the paralogs have remained distinct since their initial duplication and have steadily diverged between species. Specific CEACAM domain sequences generally follow this pattern (*Figure 4B*, *Figure 4—figure supplements 3–6*). However, the N-domains of CEACAM1, CEACAM3, CEACAM5, and CEACAM6 deviate strikingly from this norm and form a single monophyletic group (hereafter called $CCM_{1356}$), albeit one with low bootstrap support (*Figure 4B*, *Figure 4— figure supplements 3 and 4*). Within the $CCM_{1356}$ clade we observe that rather than clustering by paralog, N-domains are split into subclades representing the three major primate lineages (*Figure 4C*, *Figure 4—figure supplement 4*). In general, the close phylogenetic relationship of sequences within these clades is well supported. This topology suggests that these CEACAM N-domains are more similar to paralogous domains within the same species or primate lineage than they are to their respective orthologs across species. Several well-supported nodes provide further evidence that gene conversion is driving concerted evolution within the $CCM_{1356}$ clade (*Figure 4C*). Certain pairs of N-domains, such as CEACAM3 and CEACAM1 in gorilla and CEACAM1 and CEACAM5 in orangutan, form monophyletic groups with strong bootstrap support. As these relationships are not observed for the other domains of these CEACAM proteins, this suggests conversion events affecting only the N-domains of these CEACAMs occurred in these species. New World monkeys provide the most striking phylogenetic evidence of gene conversion among primates. For each of the four New World monkey species examined, the N-domains of CEACAM1, CEACAM5, and CEACAM6 are all more closely related within species than to their orthologs in other species, suggesting gene conversion has independently acted on the N-domains of these three CEACAMs at least four times within this single clade (*Figure 4C*). These findings are consistent with N-domains of CEACAMs 1, 3, 5, and 6 undergoing widespread concerted evolution, likely facilitated by gene conversion.

To further test for evidence of gene conversion acting on primate CEACAM family members, we applied the GARD algorithm from the HyPhy software package. GARD detects topological changes between trees inferred from segments of a gene alignment, assesses the likelihood they are consistent with recombination, and identifies potential breakpoints. Consistent with our phylogenetic examination of CEACAM homologs, GARD detects strong evidence of recombination for CEACAM1, CEACAM3, CEACAM5, and CEACAM6 (*Figure 4D*, *Figure 2—figure supplement 1A*). In all cases, breakpoints were identified at the C-terminus of the N-domain or in immediately adjacent IgC domains. This pattern is consistent with repeated N-domain gene conversion between $CCM_{1356}$ paralogs and is also in line with our phylogenetic reconstructions of CEACAM IgC domains (*Figure 4—figure supplement 5*). In addition to CEACAM1, CEACAM3, CEACAM5, and CEACAM6, GARD also indicates a recombination breakpoint for CEACAM7 that would encompass the N-domain. While we do not detect discordance in our N-domain gene tree that implicates gene conversion involving CEACAM7, there is a single instance in the IgC domain tree of a gorilla CEACAM5 IgC domain grouping more closely with homologs of the IgC domain of CEACAM7 (*Figure 4—figure supplement 5*). A breakpoint in this region is also consistent with CEACAMs with rapid N-domain evolution being involved in gene conversion events as well as previous analyses (*Zid and Drouin, 2013*). Together these results support a model in which gene conversion between rapidly diverging CEACAMs has contributed to N-domain diversification during primate evolution.

## Rapidly evolving regions of CEACAM1 are sufficient to block bacterial adhesin recognition

Phylogenetic analyses confirm that the bonobo CEACAM1 N-domain is not closely related to other primate CEACAM1 sequences but fail to strongly support its relationship to any other single CEACAM N-domain. Reasoning that the extant bonobo CEACAM1 gene may have arisen from multiple iterative recombination events, we performed a BLAST search of genomes on the NCBI database using base

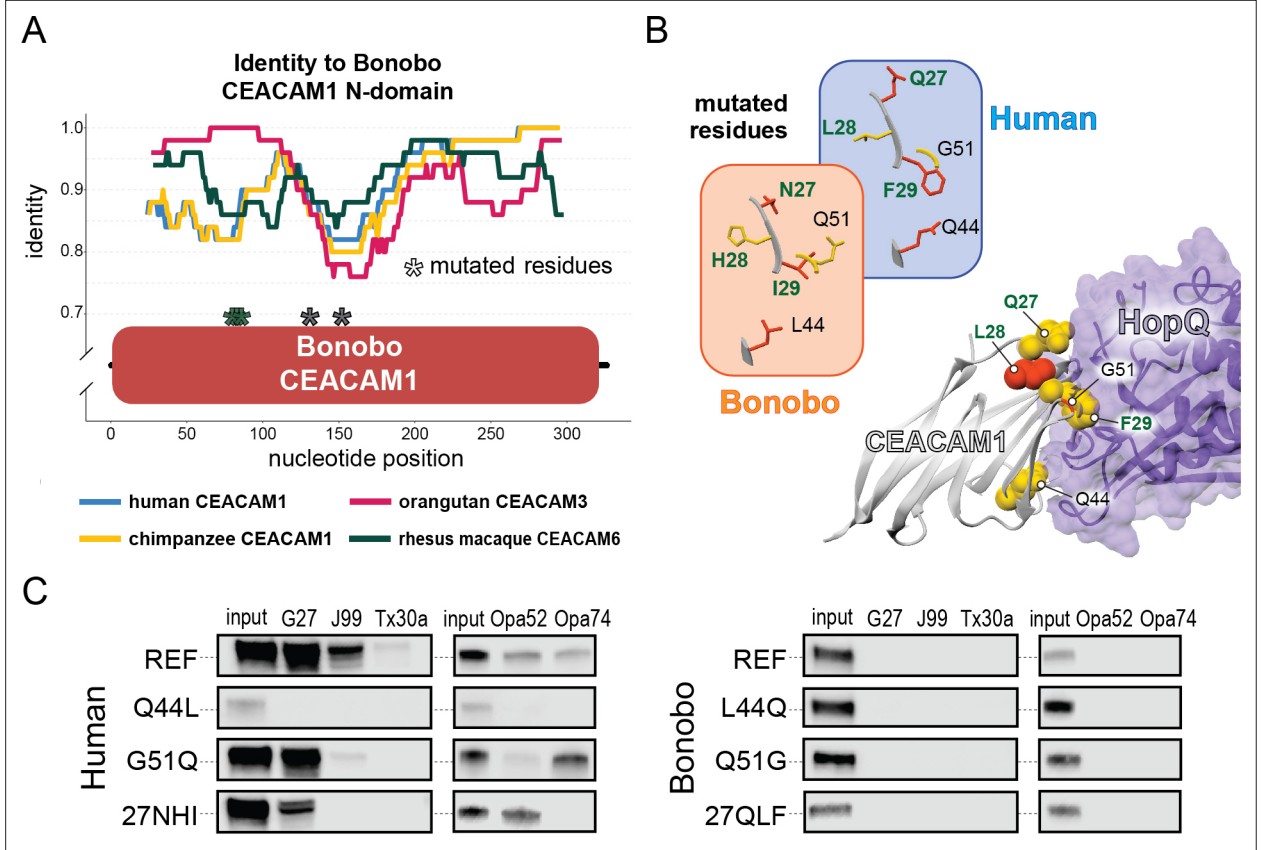

**Figure 5.** Rapid divergence of the bonobo carcinoembryonic antigen-related cell adhesion molecule 1 (CEACAM1) N-domain impairs bacterial adhesin recognition. (**A**) Graph shows a fifty base pair sliding window plotting identity between bonobo CEACAM1 N-domain sequence and other CEACAM sequences. Asterisks mark locations of residues mutated for adhesin-binding assays. (**B**) Windows show amino acids and their structures at sites selected for mutational analysis in humans and bonobos. Lower right depicts a protein co-crystal structure of human CEACAM1 and *Helicobacter pylori* G27 HopQ with sites selected for mutagenesis highlighted. (**C**) Representative western blots of pulldown experiments assaying binding between chimeric human and bonobo CEACAM1 N-domain constructs and bacterial strains.

The online version of this article includes the following source data and figure supplement(s) for figure 5:

**Source data 1.** Raw and labeled western blot images for *Figure 5C*.

**Figure supplement 1.** Alignment of rapidly evolving N-domain region in hominids.

pairs 103–303 of the bonobo CEACAM1 sequence (corresponding to resides 1–67 of the N-domain) as our query. Human and chimpanzee are roughly 86% identical to bonobo CEACAM1 in this region versus 99% identical (a single nucleotide change) in the remaining 120 base pairs (*Figure 4—figure supplement 1*). This search identifies orangutan CEACAM3 as the closest match. While the similarity between the first 120 bp of bonobo CEACAM1 and orangutan CEACAM3 is striking and the final third of the nucleotide sequence is nearly identical to human and chimpanzee CEACAM1, other segments of bonobo CEACAM1 are still quite divergent from all other N-domain sequences (*Figure 5A*). A BLAST search of this region in bonobo CEACAM1 (base pairs 221–380) indicates the greatest similarity is with the analogous region from rhesus macaque CEACAM6. However, the increased similarity of macaque CEACAM6 in this region compared to other CEACAMs is marginal (*Figure 5A*).

The extreme divergence of the bonobo CEACAM1 N-domain from other CEACAM1 homologs in even its closest relatives could indicate that this particular sequence has been evolving independently of other N-domain alleles for a long period of time as a result of balancing selection. This has been observed for other genes involved in host-pathogen conflicts, most notably major histocompatibility complex alleles (*Meyer et al., 2018*). In this case, we might expect to identify alleles similar to bonobo CEACAM1 currently circulating in other hominid populations, and likewise alleles similar to CEACAM1 sequences observed in humans and chimpanzees may be found in the larger bonobo population. In a search of human genetic variation data available through the International Genome Sample Resource

(IGSR) accessed through the Ensembl webserver (https://www.ensembl.org), there is no evidence for any alleles with similarity to bonobo CEACAM1 circulating within human populations. Searching population data from the Great Apes Genome Project (*Prado-Martinez et al., 2013*), alleles similar to bonobo CEACAM1 are not found for chimpanzees, gorillas, or orangutans. Likewise, CEACAM1 alleles similar to those found in humans and chimpanzees are not observed in any of the bonobo genomes from the same dataset. Given the information at hand, it is difficult to precisely determine the series of mutational events that produced the bonobo CEACAM1 allele or determine the likely origin point of this allele in the diversification of hominids. However, these results are consistent with multiple independent instances of gene conversion giving rise to bonobo CEACAM1, with subsequent fixation of this haplotype in bonobo populations since their divergence from chimpanzees over the last million years.

Given the large number of residue changes between human and bonobo CEACAM1, we next sought to determine if a subset of rapidly evolving sites are sufficient to either impair or restore recognition by bacterial adhesins. To test this, we generated CEACAM1 N-domain proteins in which a subset of residues between humans and bonobos were swapped. We focused on sites that are identical in humans and chimpanzees but differ in bonobos and which exhibit high $\omega$ across primates, resulting in a total of five tested sites (*Figure 5A and B*). Of these residues we chose to mutate adjacent amino acids 27–29 as a single group. This patch of sites is highly variable among the rapidly evolving CEACAMs, particularly CEACAM1, CEACAM3, and CEACAM5 (*Figure 5—figure supplement 1*). None of the 'humanized' mutants in the bonobo CEACAM1 background were sufficient to confer binding (*Figure 5C*). In contrast, introduction of bonobo residue 44 into human CEACAM1 (mutation Q44L) prevents binding by *H. pylori* and Opa-expressing strains, while introduction of bonobo variable sites 27–29 abolishes binding to Opa74 (*Figure 5C*). Mutation G51Q has no appreciable impact on binding by *H. pylori* strain G27 or *Neisseria* Opa74, but blocks binding by *H. pylori* strain Tx30a and reduces binding to J99 and Opa52. Collectively these results reveal that multiple single positions in human CEACAM1 exhibiting signatures of positive selection are sufficient to impair recognition by multiple bacterial adhesins. Moreover, these findings also demonstrate how instances of gene conversion between CEACAM paralogs could serve as large-effect adaptive mutations mediating microbial evasion.

## Abundant human CEACAM1 polymorphisms impair bacterial recognition

Pervasive evidence of positive selection acting on CEACAMs in primates raises the question as to whether CEACAM variants that evade pathogen recognition are currently segregating in human populations. To explore the existence of human CEACAM variants and their consequences for bacterial interactions, we queried human single nucleotide polymorphism (SNP) and haplotype data for rapidly evolving CEACAM paralogs available from the IGSR accessed through the Ensembl genome browser (see Materials and methods). We found that variation in the N-domains of CEACAM6, CEACAM7, and CEACAM8 predominantly consists of polymorphisms not shared with other CEACAM proteins and found on isolated haplotypes. In contrast, CEACAM1, CEACAM3, and CEACAM5 N-domain variation is composed primarily of extended haplotypes (*Figure 6—figure supplements 1–4*). Furthermore, these extended haplotypes increase similarity between CEACAM1, CEACAM3, and CEACAM5, consistent with possible gene conversion events. Indeed, some haplotypes not only have changes at nonsynonymous sites that increase similarity with these CEACAMs, but also include shared synonymous changes. These observations suggest that gene conversion among CEACAMs has occurred relatively recently and may be ongoing in human populations.

A search of polymorphisms for CEACAM1 in human populations reveals three high-frequency nonsynonymous variants within the N-domain: Q1K (rs8111171), A49V (rs8110904), and Q89H (rs8111468) (*Figure 6A*). The haplotype containing all three alternative alleles is the most frequent non-reference CEACAM1 haplotype annotated, occurring in 14% of the human population overall and in up to 43% of individuals in African populations (*Figure 6A*). In total, nearly 17% of all sequenced individuals carry at least one of these high-frequency SNPs (*Figure 6—figure supplement 2*). Of the three variants, A49V and Q89H both lie within regions of CEACAM1 known to interact with bacterial adhesins suggesting they may alter bacterial adherence (*Figure 6B*). To determine if these high-frequency CEACAM1 polymorphisms affect bacterial recognition, we generated recombinant

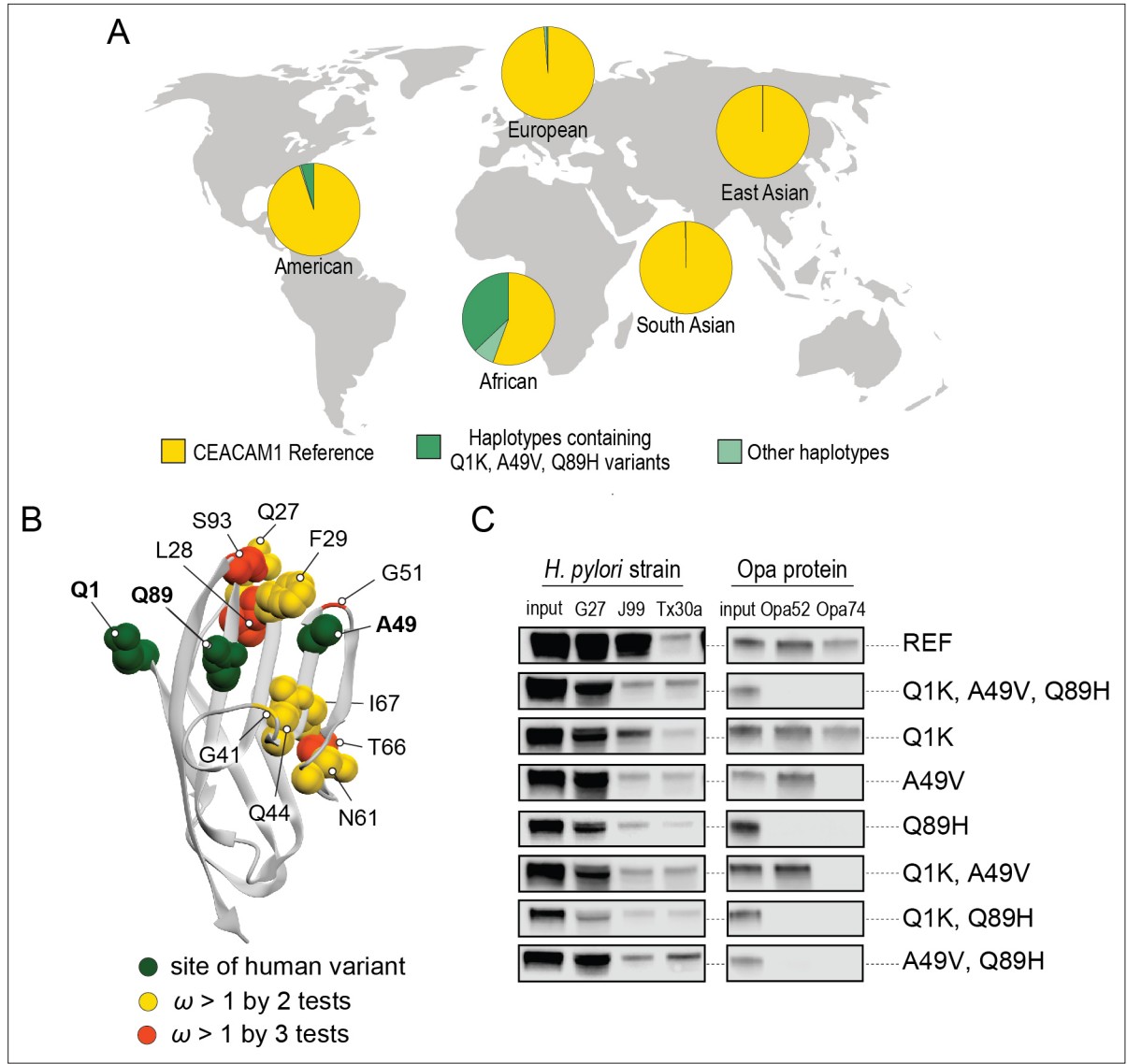

**Figure 6.** Abundant human carcinoembryonic antigen-related cell adhesion molecule 1 (CEACAM1) variants restrict pathogen binding. (**A**) Frequency of haplotypes containing variants Q1K, A49V, and Q89H across human populations (map from BioRender.com). (**B**) CEACAM1 crystal structure highlighting high-frequency human variants and sites found to be evolving under positive selection across simian primates. (**C**) Representative western blots of pulldown experiments testing binding between combinations of high-frequency human variants in the human CEACAM1 reference background and bacterial strains.

The online version of this article includes the following source data, source code, and figure supplement(s) for figure 6:

**Source code 1.** Code for analyzing carcinoembryonic antigen-related cell adhesion molecule 1 (CEACAM1) haplotypes and generating graphs for *Figure 6A*.

**Source data 1.** Data files for carcinoembryonic antigen-related cell adhesion molecule 1 (CEACAM1) haplotypes for *Figure 6A* and *Figure 6—figure supplements 1 and 2*.

**Source data 2.** Raw and labeled western blot images for *Figure 6C*.

**Figure supplement 1.** Human carcinoembryonic antigen-related cell adhesion molecule (CEACAM)-like CEACAM1 haplotypes.

**Figure supplement 2.** Human carcinoembryonic antigen-related cell adhesion molecule 1 (CEACAM1) haplotype frequencies.

**Figure supplement 2—source code 1.** Code for analyzing carcinoembryonic antigen-related cell adhesion molecule 1 (CEACAM1) haplotypes and generating graphs for *Figure 6—figure supplement 2*.

**Figure supplement 3.** Human carcinoembryonic antigen-related cell adhesion molecule 3 (CEACAM3) variation.

**Figure supplement 3—source code 1.** Code for analyzing carcinoembryonic antigen-related cell adhesion molecule 3 (CEACAM3) haplotypes and generating graphs for *Figure 6—figure supplement 3*.

*Figure 6 continued on next page*

*Figure 6 continued*

**Figure supplement 3—source data 1.** Data files for carcinoembryonic antigen-related cell adhesion molecule 3 (CEACAM3) haplotypes for *Figure 6— figure supplement 3*.

**Figure supplement 4.** Human carcinoembryonic antigen-related cell adhesion molecule 5 (CEACAM5) variation.

**Figure supplement 4—source code 1.** Code for analyzing and generating graphs for carcinoembryonic antigen-related cell adhesion molecule 5 (CEACAM5) haplotypes for *Figure 6—figure supplement 4*.

**Figure supplement 4—source data 1.** Data files for carcinoembryonic antigen-related cell adhesion molecule 5 (CEACAM5) haplotypes for *Figure 6— figure supplement 4*.

CEACAM1 N-domain variant proteins for use in our adhesin-binding assays. None of the variants are able to abolish CEACAM1 binding to our panel of *H. pylori* strains (*Figure 6C*). In contrast, *Neisseria* Opa-expressing strains exhibit highly variable recognition of multiple human CEACAM1 variants. The Q1K mutation alone has no impact on binding, while A49V abolishes recognition by Opa74, and variant Q89H abrogates binding to both Opa52 and Opa74 (*Figure 6C*). Combinatorial CEACAM1 variants reveal that these mutations behave in a dominant manner, with Q89H dominant over A49V (*Figure 6C*). Together these results demonstrate that high-frequency human polymorphisms in CEACAM1 are sufficient to impair binding by specific classes of bacterial adhesins present in human pathogens. These findings further suggest that high-frequency CEACAM variants could alter human colonization or infection by pathogenic *Neisseria*, including causative agents of gonorrhea and meningitis.

## Discussion

Our investigation of species-specific bacterial adherence to CEACAM1 revealed an unforeseen example of extreme genetic divergence within the great apes. The bonobo CEACAM1 gene could represent a rapid succession of single residue changes combined with multiple recombination events arising in bonobos under strong selection and/or a population bottleneck. Alternatively, this allele may be ancient and have been subject to balancing selection or incomplete lineage sorting in ancestral hominid populations. We also considered that the source of the bonobo CEACAM1 sequence may not be from functional CEACAM genes, but a pseudogenized CEACAM sequence or a pregnancy-specific glycoprotein, a family of proteins closely related to CEACAMs. However, a BLAST search of relevant NCBI databases (see Materials and methods) fails to identify any new genomic regions in bonobos or other primates with greater sequence identity than what had already been found. While there are multiple possible explanations for the highly divergent nature of bonobo CEACAM1, absent further evidence the origin of this particular allele remains obscure. What is clear from the example of bonobo CEACAM1, however, is the extent to which gene conversion can rapidly generate diversity between closely related species and the impact of such variation on interactions with microbes.

During the course of investigating the origin of the bonobo CEACAM1 sequence, we discovered evidence that gene conversion has shaped the evolution of many CEACAMs across primates, primarily impacting homologous proteins targeted by bacteria. While we identify several instances of likely gene conversion, results from phylogenetic analyses probably represent an underestimate of the true number of recombination events that have occurred among rapidly evolving CEACAMs in primates. Repeated episodes of gene conversion can obscure past instances of recombination and hinder their identification by gene-species tree discordance. GARD analyses and recombination detection programs in general also tend to miss many recombination events (*Bay and Bielawski, 2011*; *Kosakovsky Pond et al., 2006*). One particularly interesting example in orangutans implicates multiple conversion events impacting CEACAM1, CEACAM5, and CEACAM8 (*Figure 5—figure supplement 1*). Phylogenetic analyses indicate a species-specific conversion event between CEACAM1 and CEACAM5 in orangutans. Prior to the CEACAM1-CEACAM5 conversion, however, residues 29–64 in either CEACAM1 or CEACAM5 were likely replaced by the homologous sequence from CEACAM8. Evidence for this event includes not only multiple nonsynonymous substitutions shared with orangutan CEACAM8, but a shared synonymous substitution in both orangutan CEACAM1 and CEACAM5 only observed in hominid CEACAM8 homologs. Despite this evidence, neither our phylogenetic analyses nor GARD analyses suggest CEACAM8 has been involved in gene conversion. Like CEACAM7, the involvement of CEACAM8 in intra-paralog gene conversion is consistent with CEACAMs with rapid

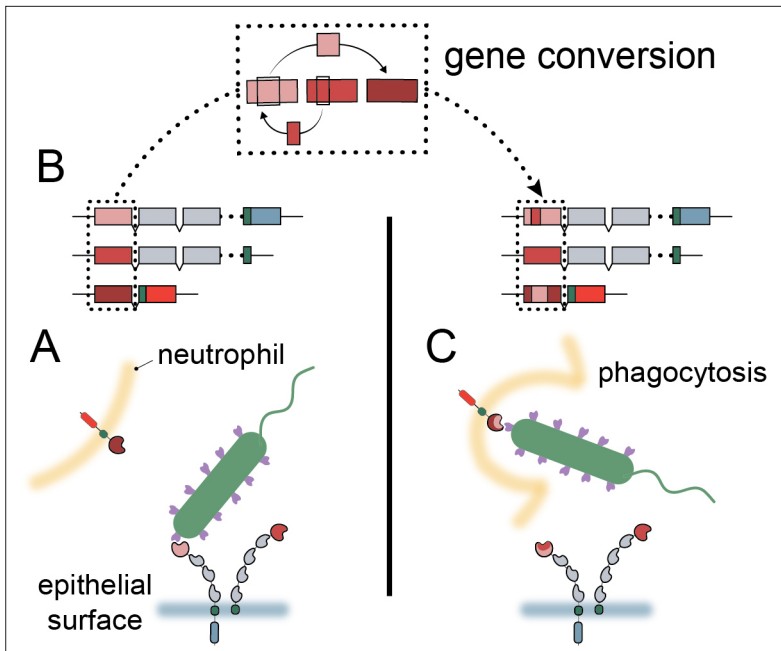

**Figure 7.** Model of carcinoembryonic antigen-related cell adhesion molecule (CEACAM) evolution in primates. (**A**) Bacterial adhesins recognize a subset of epithelial CEACAM proteins and avoid binding with decoy CEACAM receptors present on neutrophils. (**B**) Gene conversion facilitates the shuffling of regions of the CEACAM N-domain that alter binding to bacterial adhesins. (**C**) Through gene conversion outlined in B, epithelial CEACAM proteins avoid binding by bacterial adhesins while the CEACAM decoy receptor gains binding, triggering bacterial clearance through phagocytosis.

N-domain evolution participating in gene conversion events. Overall, the rapid shuffling of genetic variation among CEACAM genes that we observe could greatly augment the potential for host adaptation in the face of microbial antagonism.

It has been suggested that gene conversion between CEACAM paralogs preserves the ability of CEACAM3 to effectively mimic bacterially antagonized CEACAMs and thereby maintain its function as a decoy receptor (*Zid and Drouin, 2013*; *Zimmermann, 2019*). Indeed, our results and those of *Adrian et al., 2019*, support the importance of maintaining the similarity of CEACAM3 to other adhesin-binding CEACAMs in apes and Old World monkeys. However, our findings suggest that gene conversion does not only serve to maintain CEACAM3's mimicry function. There is no evidence that New World monkeys encode a CEACAM3 homolog, yet within this group gene conversion appears to be rampant between CEACAM1, CEACAM5, and CEACAM6 (*Figure 4C*). Additionally, we observe multiple conversion events in hominids that do not involve CEACAM3 (*Figure 4C*, *Figure 5—figure supplement 1*, and *Figure 6—figure supplement 4*). Though we cannot rule out the possibility that New World monkey CEACAM1, CEACAM5, or CEACAM6 may have evolved a mimicry function similar to CEACAM3, our observations in hominids suggest additional drivers for pervasive gene conversion within the CCM$_{1356}$ group.

High sequence similarity and close chromosomal proximity, such as that seen for the N-domains of the CCM$_{1356}$ group (*Figure 2—figure supplement 1B*, *Figure 4—figure supplement 3*), could result in periodic interconversion of paralogous sequences, which in many cases may be functionally neutral (*Bittihn and Tsimring, 2017*; *Zid and Drouin, 2013*). However, we show that the common CEACAM1 variant, Q89H, which matches CEACAM5 at this position, abolishes binding to at least two Opa proteins (*Figure 6C*). Human CEACAM1 variant A49V, which reduces binding to Opa74, also matches the equivalent residue in CEACAM5 as well as CEACAM3. These examples illustrate the potential benefit of gene conversion between epithelial CEACAM paralogs beyond CEACAM3 mimicry. Given that gene conversion within this family appears to be enriched specifically among CEACAMs recognized by bacteria, we propose that gene conversion among epithelial CEACAMs reflects a general mechanism of pathogen evasion (*Figure 7*). Such a mechanism allows beneficial sets of mutations

to spread, whether between decoys and targets or among antagonized epithelial CEACAMs, more rapidly than conversion of residues through independent mutational events (*Bittihn and Tsimring, 2017*). In addition, recombination provides decoys the ability to gain binding to CEACAM antagonists through exchanges from epithelial CEACAMs as observed for CEACAM3 (*Adrian et al., 2019*). Finally, the interchangeability of CEACAM domains could further accelerate the exchange and spread of beneficial mutations in host genomes (*Bittihn and Tsimring, 2017*). In these ways gene conversion provides an important mechanism by which the host can keep pace with rapidly evolving pathogenic microbes.

Our study examined the influence of CEACAM1 diversity on bacterial binding using adhesins from *H. pylori* and *N. gonorrhoeae* as models. However, a diverse range of pathogenic and commensal microbes are known to bind host CEACAMs, and it is unlikely that natural selection across primates has been driven by any single pathogen. What is clear from these results is the major influence that CEACAM divergence and polymorphisms can have on recognition by structurally unrelated bacterial adhesins, and moreover the impact that such diversity could have on shaping future disease outbreaks. It is also notable that recombination between paralogs contributes to the immense diversity of Opa genes *Neisseria* (*Sadarangani et al., 2011*). In this regard CEACAM-Opa interactions could reflect a general mechanism by which gene conversion between paralogs contributes to reciprocal adaptation in both hosts and microbes.

In addition to exploring the role of CEACAM gene conversion among primates, we provide evidence that this process continues to shape CEACAM diversity within human populations. The three human CEACAM1 variants we test in our adhesin-binding assay are part of a group of related CEACAM1 haplotypes that increase sequence similarity to CEACAM3 and/or CEACAM5 (*Figure 6— figure supplement 1*). Extended haplotypes that increase similarly to CEACAM1 at both synonymous and nonsynonymous positions in the N-domain are also found for CEACAM3 and CEACAM5 in humans (*Figure 6—figure supplement 3* and *Figure 6—figure supplement 4*). Indeed, haplotypes consisting of variants of putative recombination events are the most common non-reference alleles for CEACAM1, CEACAM3, and CEACAM5 (*Figure 6*, *Figure 6—figure supplements 1–4*). Variant sites in these proteins tend to lie along the protein-binding interface of the N-domain and often impact residues known to influence adhesin recognition. The relationships between these different CEACAM haplotypes appears to be complex, as many different combinations of partial variant haplotypes exist for each CEACAM paralog. The haplotype structures we observe suggest these CEACAM variants are the result of one or more conversion events between paralogous sequences, likely followed by further recombination with the major CEACAM allele.

Important questions remain regarding the rapid evolution of a subset of primate CEACAM proteins. Among these questions is why CEACAM7 and CEACAM8 show similar patterns of evolution to bacterially antagonized CEACAMs despite no known instances of bacterial antagonism. The simplest explanation is that CEACAM7 and CEACAM8 are themselves the targets of as yet unidentified pathogen antagonists (*Sintsova et al., 2015*). Alternatively, their rapid evolution may reflect pressure to maintain binding with rapidly evolving bacterially antagonized CEACAMs (*Gray-Owen and Blumberg, 2006*; *Skubitz and Skubitz, 2008*) could merely be a result of their genomic proximity to rapidly evolving CEACAMs prone to gene conversion (*Zid and Drouin, 2013*) or could be the result of some as yet unknown evolutionary pressures. Another intriguing aspect of rapid CEACAM evolution is the impact rapid divergence might have beyond interactions with pathogenic microbes. Given the extensive overlap of CEACAM-binding sites among unrelated bacterial adhesins, the ramifications of rapid CEACAM evolution likely extend beyond the adhesins of pathogens to those of commensal and beneficial microbes as well. For commensal microbes that rely on these interaction surfaces, CEACAM evolution could significantly alter their ability to colonize the host. The impact of CEACAM divergence on composition of the host microbiome and/or the evolution of commensal strains warrants further investigation.

Studies of other 'housekeeping' proteins targeted by pathogens have found that sites under positive selection typically do not overlap with sites involved in essential host functions (*Barber and Elde, 2014*; *Demogines et al., 2013*). This is clearly not the case for CEACAMs, where we observed extensive overlap between sites involved in host protein interactions, sites targeted by bacterial adhesins, and sites undergoing rapid evolution (*Figure 2B*). How CEACAMs are able to rapidly evolve while maintaining their other essential host protein interactions remains a mystery. Future studies on

CEACAM protein functions, interaction networks, and pathogen antagonism will likely clarify these outstanding questions regarding rapidly evolving CEACAMs.

Collectively our study provides evidence that repeated adaptation among primate CEACAMs has shaped host-specific cell adherence by diverse pathogenic bacteria. We find that over half of the CEACAM paralogs found in humans display signatures of positive selection across the primate lineage, localized primarily to the extracellular N-domain. We further discovered that rapid evolution of CEACAM N-domains has been facilitated by extensive 'shuffling' of sequences between a subset of CEACAM paralogs likely through repeated gene conversion. The diversification of primate CEACAM N-domain sequences has likely had significant consequences for interactions between primates and bacteria. Consistent with observations across other primate species, we also provide evidence that gene conversion events may impact bacterial pathogen recognition of CEACAMs in contemporary human populations. Together this work reveals how dynamic evolutionary processes have shaped bacterial-host cell associations with consequences for infectious disease susceptibility.

# Materials and methods

**Key resources table**

| Reagent type (species) or resource | Designation | Source or reference | Identifiers | Additional information |
|---|---|---|---|---|
| Strain, strain background (*Helicobacter pylori*) | G27 | *Baltrus et al., 2009* | | |
| Strain, strain background (*Helicobacter pylori*) | J99 | *Alm et al., 1999* | | |
| Strain, strain background (*Helicobacter pylori*) | Tx30a | ATCC | 51932 | |
| Strain, strain background (*Helicobacter pylori*) | *omp27::cat-sacB* in NSH57 | *Yang et al., 2019* | | *H. pylori* strain G27 with HopQ deletion |
| Strain, strain background (*Escherichia coli*) | Rosetta (DE3) pLyS | Lab collection | | *E. coli* strain for outer membrane IPTG inducible expression of *Neisserial* Opa proteins |
| Strain, strain background (*Escherichia coli*) | DH5α | Lab collection | | *E. coli* strain for maintenance and propagation of pET-28a plasmid constructs |
| Strain, strain background (*Escherichia coli*) | One Shot Top10 Chemically Competent cells | Thermo Fisher Scientific | C404010 | *E. coli* strain for cloning, maintenance and propagation of pcDNA3 GFP LIC plasmid constructs |
| Cell line (*Homo sapiens*) | HEK293T | ATCC | RRID:CVCL_0063; CRL-3216 | |
| Recombinant DNA reagent | pET-28a (plasmid) | Genscript | | Plasmid backbone for expression of *Neisserial* Opa proteins |
| Recombinant DNA reagent | pcDNA3 GFP LIC (plasmid) | Addgene | RRID:Addgene_30127; #30,127 | Plasmid backbone for expression of primate CEACAM1 N-domain constructs in HEK293T cells |
| Antibody | Mouse monoclonal antibody mixture; Mouse α-GFP clones 7.1 and 13.1 | Sigma-Aldrich | RRID:AB_390913; 11814460001 | 1:10³ dilution; Primary antibody for visualization of GFP labeled CEACAM1 N-domain constructs |
| Antibody | Goat polyclonal antibody; goat α-mouse conjugated to horseradish peroxidase | Jackson ImmunoResearch | RRID:AB_10015289; 115-035-003 | 1:10⁴ dilution; Secondary antibody for visualization of GFP labeled CEACAM1 N-domain constructs |
| Other | Advansta WesternBright ECL HRP Substrate | Thomas Scientific | K-12049-D50 | Reagent to visualize proteins bound by secondary antibody in a western blot |

*Continued on next page*

*Continued*

| Reagent type (species) or resource | Designation | Source or reference | Identifiers | Additional information |
|---|---|---|---|---|
| Software, algorithm | PAML4.9h | http://abacus.gene.ucl.ac.uk/software/paml.html *Yang, 2007* | RRID:SCR_014932 | |
| Software, algorithm | FUBAR | https://www.datamonkey.org *Murrell et al., 2013* | RRID:SCR_010278 | |
| Software, algorithm | MEME | classic.datamonkey.org *Murrell et al., 2012* | RRID:SCR_010278 | |
| Software, algorithm | GARD | classic.datamonkey.org *Kosakovsky Pond et al., 2006* | RRID:SCR_010278 | |
| Sequence-based reagent | bon_gCCM1N_F3 | This paper | PCR primer | Primer for initial amplification of bonobo CEACAM1 N-domain from genomic DNA [TTCACAGAGTGCGTGTACCC] |
| Sequence-based reagent | bon_gCCM1N_R2 | This paper | PCR primer | Primer for initial amplification of bonobo CEACAM1 N-domain from genomic DNA [CCTCCCAGGTTCAAGCGATT] |
| Sequence-based reagent | bon_gCCM1N_F1 | This paper | PCR primer | Primer for secondary amplification of bonobo CEACAM1 N-domain from genomic DNA [CAGTGGAGGGGTGAAGACAC] |
| Sequence-based reagent | bon_gCCM1N_R1 | This paper | PCR primer | Primer for secondary amplification of bonobo CEACAM1 N-domain from genomic DNA [CATGTTGGTCAGGCTGGTCT] |
| Sequence-based reagent | bon_gCCM1N_seqF1 | This paper | Sequencing primer | Primer to sequence bonobo CEACAM1 N-domain amplified from genomic DNA [CCCGTTTTTCCACCCTAATGC] |
| Sequence-based reagent | bon_gCCM1N_seqF4 | This paper | Sequencing primer | Primer to sequence bonobo CEACAM1 N-domain amplified from genomic DNA [GGGGAAAGAGTGGATGGCAA] |
| Sequence-based reagent | bon_gCCM1N_seqR2 | This paper | Sequencing primer | Primer to sequence bonobo CEACAM1 N-domain amplified from genomic DNA [TGGGGGAATCACTCACGGTA] |
| Biological sample (*pan paniscus*) | AG05253 | Nels Elde | RRID:CVCL_1G37 | Bonobo genomic DNA sample |
| Software, algorithm | R v4.1.2 | https://cran.r-project.org/ | RRID:SCR_003005 | |
| Software, algorithm | Python 3.7 | Python Software Foundation https://www.python.org/ | RRID:SCR_008394 | |
| Software, algorithm | Jupyter Notebook 5.7.4 | Project Jupyter https://jupyter.org/ | RRID:SCR_018315 | |
| Software, algorithm | Anaconda Navigator 1.9.12 | Anaconda, Inc https://www.anaconda.com/ | | |

## Primate comparative genetics

### Sequence identification

Orthologs for human CEACAM genes were identified through BLAST searches of primate reference genomes available through the NCBI BLAST webserver (*Boratyn et al., 2013*). Full-length genomic regions for annotated human CEACAMs were used as query sequences. A full record of CEACAM orthologs identified and a partial record of BLAST results, including date accessed, query coverage and identity, as well as information on synteny, are listed in *Figure 2—source data 1A-C*. Orthology was established by sequence identity, reciprocal best-BLAST hit, as well as intron structure and synteny. In total, we were able to extract 186 primate CEACAM sequences for analysis. We could

not identify orthologs of every human CEACAM in every primate species, in some cases because of lineage-specific gains and losses and in some cases likely because of incomplete genome assembly. As a result, the number of primate orthologs available for evolutionary analysis and phylogenetic reconstruction for each human CEACAM range from 11 to 19 (*Figure 2—source data 1A*).

## Sequence alignment and trimming

Orthologous protein-coding sequences were extracted from CEACAM genes as follows. Multiple sequence alignments of the full-length gene were done using MAFFT alignment software as implemented in Geneious Prime 2020.2.2 with default settings. Alignments were manually corrected to correspond to human exon splice sites. Regions corresponding to human exons were then extracted, realigned, and minimally trimmed so all sequences were in-frame and orthologous codons aligned. So as not to exclude any protein-coding regions from evolutionary analysis, all human exons for a given CEACAM were concatenated and treated as a single protein-coding sequence. Consequently, representations of CEACAM proteins in figures are not necessarily indicative of mature peptides, but rather represent all parts of the CEACAM protein that could potentially have been subject to positive selection. Gaps in alignments were removed for evolutionary analyses (*Figure 2—source data 2*) but were retained for tree building (*Figure 4—source data 1*).

## CEACAM3 exons

Almost all Old World monkey CEACAM3 genes analyzed had two extra exons annotated compared to humans. These exons are located between the exon encoding the N-domain and the transmembrane domain and are predicted by InterProScan (*Quevillon et al., 2005*), as implemented in Geneious Prime 2020.2.2, to encode the IgC-like domains typical of this region of CEACAM proteins. The majority of Old World monkeys have two exons annotated and all primates, including hominids, have strongly conserved sequences in this region, though hominids all encode premature stop codons. With the exception of the second IgC exon in colobus, these exons would allow for the translation of full-length CEACAM proteins. While exon annotation differences between primate CEACAM genes is not unusual, the conservation of these sequences across primates containing a CEACAM3 gene, including in hominids where they are not annotated, is striking. To the best of our knowledge CEACAM3 transcripts for humans or other primates including either of these extra IgC domains have not been reported. Indeed, the exon closest to the N-domain likely does not encode a functional protein in most hominids as a result of a premature stop codon. However, the strong conservation of these sequences across primates could indicate these exons encode functional protein segments in at least some species. For this reason, these exons and their orthologous sequences in hominids were included in downstream evolutionary analyses.

## CEACAM5 trimming

The differences in number and likely arrangement of IgC domains in primate CEACAM5 orthologs prevented alignment of all full-length CEACAM5 genes into a single multiple sequence alignment for extracting human orthologous protein-coding sequences. Instead, sequences were first aligned in three groups: New World monkeys, leaf-eating monkeys (black-and-white colobus, black snub-nosed monkey, and golden snub-nosed monkey), and the remaining Old World monkey sequences with the hominid sequences. There were enough similarities with human exons for orthologous exon sequences to be assigned and extracted for New World monkeys and the Old World monkey/ Hominid group, but not for the leaf-eating monkeys group. For leaf-eating monkeys the predicted exons in common between species in this group were extracted. After extracting coding sequences for each group individually, the extracted sequences were then aligned in a single multiple sequence alignment. However, the large gaps caused by missing IgC sequences relative to human CEACAM5 posed a problem for evolutionary analyses which require gaps to be removed from sequences prior to analysis. We were concerned that choices made regarding which sequences were removed would unduly influence the results of evolutionary analyses or result in lower coverage of the evolutionary history of the entire coding sequence. To account for this, three strategies of trimming alignment gaps were carried out and the results of each used in separate evolutionary analyses. For the first strategy (datasets 1 and 4) every species whose sequence contained gaps corresponding to missing IgC domains was removed. These species were black-and-white colobus, black snub-nosed monkey,

golden snub-nosed monkey, drill, sooty mangabey, and common marmoset. This resulted in the longest sequence for analysis (2 kb) including six predicted IgC domains, but the smallest number of species represented (12). In the second strategy (datasets 2 and 5) primate sequences with gaps corresponding to the largest number of missing IgC domains (4) were removed, while those with only two missing domains were retained, and the alignment region containing the sequence gap caused by the missing domains removed, giving a smaller alignment (1.4 kb, with four IgC domains), but more species (16). For this strategy sooty mangabey, and common marmoset were removed from the analysis. For the third strategy (datasets 3 and 6) all species for which complete CEACAM5 gene sequences could be identified were retained and all gaps corresponding to missing IgC domains removed. This gave the smallest sequence (0.9 kb, retaining two IgC domains), but provided the largest number of represented species (18). Evolutionary analyses for these strategies are included in *Figure 2—figure supplement 1*, *Figure 2—source data 1A and B*, and *Figure 2—source data 3*.

## Alignment comparison between MAFFT and MUSCLE

To confirm that our alignment method was not biasing the assignment of orthology of coding sequences to human exons, we compared the results of alignments of extracted exons using MAFFT (*Katoh and Standley, 2013*) and the alternative program MUSCLE (*Edgar, 2004*), both as implemented in Geneious Prime 2020.2.2. With the exceptions of CEACAM7 and CEACAM5, there were no drastic changes between alignments performed using MAFFT and those done using MUSCLE. Upon inspection the discrepancy between MAFFT and MUSCLE alignments for CEACAM7 could be attributed to an approximately 7 kb insertion in the orangutan CEACAM7 gene relative to all other primates. Upon removing this insertion alignments with both MAFFT and MUSCLE were in agreement. Discrepancies between alignments of CEACAM5 with MAFFT and MUSCLE were due to differences in how the programs aligned sequences corresponding to IgC domains, likely as a result of differences in the number and possibly the arrangement of these domains between primates. MAFFT and MUSCLE alignments were carried out for each of the three different trimmed versions of CEACAM5 (see above) and each set of sequence alignments was tested using each of the evolutionary analysis methods. All other evolutionary analyses were carried out using sequences trimmed according to MAFFT alignments.

The results of CEACAM5 evolutionary analyses were largely similar regardless of which alignment or trimming method was employed, identifying similar patterns of selection (sites under selection concentrated in the N-domain) and many of the same sites under selection. Results presented in the paper are for dataset 1 (ds1) which contains the largest number of domains and using the MAFFT alignment to match the method used for other CEACAM analyses presented. Results for alternative CEACAM5 trimming and alignment methods are included in *Figure 2—figure supplement 1*, *Figure 2—source data 1A and B*, and *Figure 2—source data 3*.

## Bonobo CEACAM1 N-domain sequence verification

For initial verification of the bonobo CEACAM1 N-domain sequence, we utilized currently available bonobo genome sequence data. Reads were not available for the genome assembly from which bonobo CEACAM sequences were identified for the evolutionary analyses described here. However, a more recent assembly of a different bonobo individual became available during the course of this study and sequencing reads for this new de novo assembly were deposited (*Mao et al., 2021*). The CEACAM1 genomic region of the newer assembly was 99% identical to the older version while the coding sequences differ at only a single nucleotide outside of the N-domain. Furthermore, examining the reads used to assemble the newer genome we confirmed that multiple reads covered the length of the bonobo CEACAM1 N-domain and included the highly diverged nucleotides of the binding region in contiguous reads. Additionally, we examined CEACAM1 sequences for the 13 bonobo individuals sequenced as part of the Great Apes Genome Project (*Prado-Martinez et al., 2013*). Genomes for these individuals were constructed using a reference-based assembly method to the human genome. The assembled sequences largely supported the highly diverged N-domain seen in the reference genome; however, there was a 31 bp region that was identical to the human CEACAM1 sequence rather than the two de novo bonobo sequences. Examining reads from these individuals failed to support human sequences at this position and in fact supported the more divergent sequence seen in the bonobo de novo assemblies.

To further validate our results, we directly sequenced the bonobo CEACAM1 N-domain region by Sanger sequencing. Briefly, the nucleotide sequence of the bonobo CEACAM1 N-domain was amplified and sequenced from a sample of genomic DNA isolated from bonobo AG05253 (*Elde et al., 2009*) (a gift from Nels Elde). Products of the size expected for the region of CEACAM1 targeted in an initial PCR were isolated by size selection on a 1% agarose gel. This product was purified from the gel and used as template for further amplification of the N-domain region of bonobo CEACAM1 with a second set of primers. After gel purification this product was sequenced through GeneWiz (South Plainfield, NJ) using custom primers. Both sets of PCRs used a touchdown PCR method and an annealing temperature of 65°C.

Nucleotide BLAST searches on the NCBI webserver for bonobo N-domain sequences were performed with query sequences searching against the RefSeq Genome Database (refseq_genomes) for the organism groups 'Homo/Pan/Gorilla groups' (taxid: 207598) and 'Primates' (taxid: 9443), while excluding 'bonobos' (taxid: 9597), except in the case of searches within bonobo for potential gene conversion donor sequences.

## Identification of human CEACAM N-domain variation

Human haplotype data for CEACAM1, CEACAM3, CEACAM5, CEACAM6, CEACAM7, and CEACAM8, available through the IGSR (https://www.internationalgenome.org/) was accessed through the Ensemble genome browser (https://www.ensembl.org/). For each CEACAM the haplotypes identified for the Matched Annotation from NCBI and EMBL-EBI (MANE) Select v0.92 transcript were used (*Figure 6—source data 1*, *Figure 6—figure supplement 3—source data 1*, *Figure 6—figure supplement 4—source data 1*). All coding sequence haplotypes for the MANE Select transcript were downloaded and analyzed in excel as well as in R using custom scripts (*Figure 6—source code 1*, *Figure 6—figure supplement 2—source code 1*, *Figure 6—figure supplement 3—source code 1*, *Figure 6—figure supplement 4—source code 1*).

## Phylogenetic analyses
### PAML/FUBAR/MEME/GARD

Evolutionary analyses were performed individually for each group of human CEACAM-coding sequence orthologs (*Figure 2—source data 2*). Only CEACAM21 was excluded from evolutionary analyses, since it was found only in hominid genomes and has likely been lost in the pan lineage (*Figure 2—source data 1A and B*) resulting in only three closely related sequences being available for comparison, insufficient for robust phylogenetic-based evolutionary analysis. CEACAM21 sequences were included in subsequent phylogenetic reconstructions.

CEACAM-coding sequences and specific amino acids were tested for evidence of positive selection using the PAML NS sites program under the codon model F3 × 4 (*Yang, 2007*). A relevant primate species tree, based on primate species relationships detailed in *Pecon-Slattery, 2014*, was provided for each analysis (*Figure 2—source data 2*). To determine the likelihood a gene was evolving under positive selection, likelihood ratio tests were performed comparing the models of selection M1&M2 as well as M7&M8 (*Figure 2—source data 1D*; *Bielawski, 2013*). Sites evolving under positive selection (*Figure 2—source data 1E*) were identified by PAML using the Bayes empirical Bayes analysis as implemented in the NS sites package for evolutionary Model 2, which has been shown to be more robust to error due to recombination than the alternative, Model 8, when identifying sites under selection (*Anisimova et al., 2003*). In addition, sites under selection were identified using the HyPhy package programs FUBAR and MEME (*Murrell et al., 2013*; *Murrell et al., 2012*) as implemented on the Datamonkey web servers (https://www.datamonkey.org and classic.datamonkey.org, respectively) (*Delport et al., 2010*; *Kosakovsky Pond and Frost, 2005*; *Pond et al., 2005*; *Weaver et al., 2018*). For FUBAR and initial MEME analyses, species trees of the relevant primates were provided to inform analyses of evolution. HyPhy GARD analyses (classic.datamonkey.org) were used to identify evidence of recombination and the number and approximate locations of breakpoints (*Figure 2—source data 1E*, *Figure 2—source data 3*; *Kosakovsky Pond et al., 2006*). When GARD detects evidence of recombination the program splits sequence alignments at the predicted breakpoint and creates new phylogenies for each set of sequences. These updated GARD-informed phylogenies for CEACAM sequences were used for MEME analyses to account for errors in calling sites under selection due to recombination. Prior to running MEME and GARD analyses the 'automatic model selection tool'

provided by classic.datamonkey.org was used to determine the most appropriate model of selection under which to run analyses. For PAML, sites with posterior probability >0.95 were considered to have strong support to be evolving under positive selection (*Yang et al., 2005*), while >0.9 posterior probability supported sites found by FUBAR (*Murrell et al., 2013*), and p-values ≤ 0.05 supported sites found by MEME (*Murrell et al., 2012*). Results files for evolutionary analyses (*Figure 2—source data 3*) were analyzed and visualized using custom R (*R Development Core Team, 2019*) and python scripts (*Figure 2—source code 1*, *Figure 2—figure supplement 1—source code 1* , *Figure 4—source code 1*). Python code was run in a Jupyter notebook launched from the Anaconda Navigator.

## Tree building
Phylogenetic trees were constructed using our panel of primate CEACAM-coding sequences identified as described above. Multiple sequence alignments on which tree constructions were based were done by translation alignment using default settings of the MAFFT sequence alignment software as implemented in Geneious Prime 2020.2.2. For domain-specific phylogenetic reconstruction domains were identified using InterProScan (*Quevillon et al., 2005*) in Geneious Prime. Assignments for immunoglobulin-like domains, that is, the IgV-like (N-domain) and IgC domains were based on predictions by the Superfamily database (*Wilson et al., 2009*) and cytoplasmic domain assignments were based on the PHOBIUS database (*Käll et al., 2004*). Transmembrane domains were excluded from analyses due to their particularly small sequence length, which can make tree building unreliable due to limited phylogenetically informative sites. Indeed, relatively short sequence lengths for the other domains, typically around 300 bp or less, along with often high sequence similarity likely decreased the reliability and statistical support for our domain trees. However, even with these limitations in many cases relationships between domains were resolved with high bootstrap support, particularly for peripheral nodes and clades and for CEACAMs not found to be evolving rapidly. Phylogenetic reconstructions of sequence alignments (*Figure 4—source data 1*) were done using the PhyML 3.0 web browser (http://www.atgc-montpellier.fr/phyml/) with default settings and confidence testing by 1000 bootstrap replicates (*Guindon et al., 2010*).

## Data visualization
Visualization of evolutionary analyses, phylogenetic trees, sequence identity, and haplotype frequencies was done in R (*R Development Core Team, 2019*) using the R packages BiocManager (*Morgan, 2019*), treeio (*Wang et al., 2020*), ggplot2 (*Wickham, 2016*), ggtree (*Yu et al., 2018*; *Yu et al., 2017*), evobiR (*Blackmon and Adams, 2015*), and ggforce (*Pedersen, 2021*). Protein structures were visualized using the UCSF Chimera package version 1.13.1. Chimera is developed by the Resource for Biocomputing, Visualization, and Informatics at the University of California, San Francisco (supported by NIGMS P41-GM103311) (*Pettersen et al., 2004*).

## CEACAM1-binding assays
### CEACAM1 N-domain assembly PCR for expression plasmid construction
Oligonucleotides were designed to assemble expression cassettes containing the human IgκK signal sequence followed by a primate CEACAM1 N-terminal domain, and finally a STREPII tag and tobacco etch virus (TEV) protease site. Templates for primate CEACAM1, Igκ, and the STREPII tag/TEV protease site were synthesized commercially (Genscript & Integrated DNA Technologies, Piscataway, NJ, USA). Oligonucleotides were designed to amplify each fragment with approximately 20 bp of template overlap for PCR assembly as detailed in *Supplementary file 1A and B*. Individual PCR fragments for Igκ, the N-terminal domain of primate CEACAM1, and the STREPII/TEV tag were generated using the Phusion (Finnzymes) high-fidelity polymerase and were subsequently gel-purified (Gel DNA Extraction Kit, Zymo Research). For assembly, 0.2 µL of each purified PCR fragment was added as template in a Phusion PCR reaction with the outermost primer set (Igκ pCDNA GFP LIC F and GFP STREP II CEACAM1N R3). The assembly PCR was separated on a 1% agarose gel and the band corresponding to the correct assembly length was gel extracted and purified.

### LIC cloning in pcDNA3 GFP LIC vector
The pcDNA3 GFP LIC vector (a gift from Scott Gradia; Addgene plasmid ##30127) is an empty LIC vector derived from pcDNA3.1(+) which adds a C-terminal GFP gene to the protein of interest. For

vector preparation, the pcDNA3 GFP LIC vector was linearized with SspI (New England Biolabs) for 1 hr at 37°C and then purified (DNA Clean-up and Concentration, Zymo Research). The linearized vector was treated with T4-DNA Polymerase (New England Biolabs) and dGTP as per protocol (https://qb3. berkeley.edu/facility/qb3-macrolab/projects/lic-cloning-protocol/) and incubated at 22°C for 40 min. The enzyme was then heat-denatured at 75°C for 20 min. For insert preparation, purified products from the assembly PCR were similarly treated with T4 DNA Polymerase and dCTP per protocol. The annealing reaction was performed by combining 3 μL of treated vector, and 3 μL of treated PCR fragment in 20 μL total volume for 30 min at room temperature. Six μL of the annealing reaction were transformed into One Shot Top10 Chemically Competent cells per manufacturer's instructions (Thermo Fisher Scientific). Single colonies were prepped (ZymoPURE Plasmid Miniprep Kit, Zymo Research) and proper insertion was confirmed by Sanger sequencing (GeneWiz, South Plainfield, NJ).

## Introduction of variant residues into CEACAM1 constructs

Mutations to introduce bonobo CEACAM1 residues and population variants into the human CEACAM1 reference as well human CEACAM1 residues into the bonobo CEACAM1 reference sequence were done by site-directed mutagenesis using mutation-specific primers designed using the Agilent Quik-Change Primer Design tool (https://www.chem.agilent.com/store/primerDesignProgram.jsp), then transformed into One Shot Top10 chemically competent cells for amplification and sequence verification. Plasmids were extracted for further use using the ZymoPURE II Plasmid Maxiprep kit.

## Protein expression

Recombinant CEACAM1 expression plasmids were transfected into human HEK293T cells using the Lipofectamine 3000 transfection kit following manufacturer's instructions. Two days post transfection cell supernatant was collected and filter sterilized and cells were collected and lysed. Expression of proteins was confirmed by western blotting. The HEK293T cell line used in this study was obtained from ATCC which authenticated and screened the line for mycoplasma prior to shipping.

## Bacterial strains and culture

*H. pylori* strains G27 (*Baltrus et al., 2009*), J99 (*Alm et al., 1999*), Tx30a (ATCC 51932), and the G27 HopQ deletion strain (*omp27::cat-sacB* in NSH57) (*Yang et al., 2019*) were cultured microaerobically at 37°C on Columbia agar plates supplemented with 5% horse blood 0.2% beta cyclodextrin, 0.01% amphotericin B, and 0.02% vancomycin. To assay binding between recombinant primate CEACAM1 N-domain proteins and *H. pylori* strains, *H. pylori* strains were grown for 2–5 days on solid media, collected, and suspended in Brain Heart Infusion Media; 500 μL of bacterial suspension were then incubated with 100 μL of CEACAM protein for 30 min, rotating on a nutator. Bacteria were then washed twice with cold PBS. Samples to be visualized by western blotting were pelleted and resuspended in 1× Laemmli buffer. Samples to be examined by flow cytometry were suspended in 0.5–1 mL of PBS.

The use of *E. coli* to express MS11 and VP1 *N. gonorrhoeae* Opa proteins was described previously (*Roth et al., 2013*). For this project plasmids expressing Opa proteins, Opa52 (*Kupsch et al., 1993*) and Opa74 (*Roth et al., 2013*), were synthesized in the pET-28a vector background by GeneScript. Synthesizing Opa expression plasmids bypassed the subcloning described in previous works that allowed outer membrane expression, so an N-terminal signal sequence from the OMP A protein, native to the outer membrane of *E. coli*, was added by the manufacturer to express Opa proteins on the outer membrane of *E. coli*. NcoI and HindIII restriction sites were used to add OMP A and Opa sequences to the pET-28a plasmid. Opa expression vectors were transformed into *E. coli* DH5α cells for maintenance, replication, and sequence verification. Plasmids were extracted for further use using the Zymo Research Zyppy Plasmid miniprep kit. For pulldown experiments Opa expression plasmids were transformed into Rosetta (DE3) pLysS *E. coli* cells to allow for inducible expression of Opa proteins. Cells were grown to an optical density of $OD_{600}$ 0.4–0.6, then IPTG (isopropyl β-d-1-thiogalactopyranoside) was added to a concentration of 100 mM to induce expression of Opa proteins. Bacterial cells were left to induce for 3 hr at 37°C. For pulldown assays 300 μL of induced *E. coli* cell culture was incubated with 100 μL of CEACAM1 protein construct as processed as described for *H. pylori*. All *E. coli* cells were cultured at 37°C in LB (Luria-Bertani) broth.

## Western blotting and flow cytometry

Pulldown experiments assayed by western blotting were visualized using a commercially available mixture of Mouse α-GFP clones 7.1 and 13.1 (Sigma-Aldrich) at a dilution of $1:10^3$ for the primary antibody incubation followed by secondary incubation with goat α-mouse conjugated to horseradish peroxidase ($1:10^4$ dilution) (Jackson ImmunoResearch) and visualized by WesternBright ECL HRP Substrate (Thomas Scientific). For pulldowns visualized by western blotting, CEACAM1 protein input samples were prepared by mixing 20 μL of protein with 20 μL 2× Laemmli then boiled at 95°C for 5 min and centrifuged at max speed for 5 min, before visualization by western blotting alongside pulldown samples. GFP fluorescence of primate CEACAM1 constructs bound to *H. pylori* strain G27 was also measured by flow cytometry, with 10,000 events per sample measured. *H. pylori* strain G27 incubated alone was used as a negative control. Flow cytometry data was analyzed using FlowJo v10.5.3. Western blots and flow cytometry experiments depicted are representative of at least three independent replicates performed at different times.

## Acknowledgements

We are grateful to members of the Barber lab, especially Caitlin Kowalski, for helpful discussions and feedback on the manuscript. This work was supported by National Institutes of Health grants R35GM133652 (to MFB) and F32AI147565 (to EPB). We thank Karen Guillemin for strains and assistance with *H. pylori* as well as Nina Salama for sharing *H. pylori* strains. We thank Andrew Kern and CJ Battey for helpful discussions and assistance with population genetic analyses. We thank Nels Elde for the bonobo AG05253 genomic DNA sample as well as thoughtful discussions related to this work. We thank Jacob Laser for early assistance with CEACAM phylogenetics. We also thank Scott Gradia for the gift of vector pcDNA3 GFP LIC.

## Additional information

### Funding

| Funder | Grant reference number | Author |
| --- | --- | --- |
| National Institutes of Health | R35GM133652 | Matthew F Barber |
| National Institutes of Health | F32AI147565 | EmilyClare P Baker |

The funders had no role in study design, data collection and interpretation, or the decision to submit the work for publication.

### Author contributions

EmilyClare P Baker, Conceptualization, Formal analysis, Investigation, Validation, Visualization, Writing – original draft, Writing – review and editing, Methodology; Ryan Sayegh, Kristin M Kohler, Investigation, Methodology, Writing – review and editing; Wyatt Borman, Claire K Goodfellow, Eden R Brush, Investigation, Writing – review and editing; Matthew F Barber, Conceptualization, Funding acquisition, Investigation, Methodology, Project administration, Resources, Supervision, Writing – review and editing, Writing – original draft

### Author ORCIDs

EmilyClare P Baker (ID) http://orcid.org/0000-0002-7875-2144
Matthew F Barber (ID) http://orcid.org/0000-0003-2008-2165

### Decision letter and Author response

Decision letter https://doi.org/10.7554/eLife.73330.sa1
Author response https://doi.org/10.7554/eLife.73330.sa2

# Additional files

## Supplementary files

• Supplementary file 1. A. Oligomers and DNA templates. Table of oligomers, DNA templates, and their order in assembly reactions used to assemble carcinoembryonic antigen-associated cell adhesion molecule 1 (CEACAM1) N-domain expression plasmids. B. Sources templates for plasmid components. Table listing sources of template sequences for CEACAM1 and other plasmid components used for expression plasmid construction.

• Transparent reporting form

## Data availability

The following files contain the images, data and/or code used to perform analyses and generate figures for this work, Figure 3 - Source data 1, Figure 3 - SuppFig 1 - source data 1, Figure 5 - Source data 1, Figure 6 - Source data 1, Figure 6 - Source data 2, Figure 6 - SuppFig 3 - source data 1, Figure 6 - SuppFig 4 - source data 1.

The following previously published datasets were used:

| Author(s) | Year | Dataset title | Dataset URL | Database and Identifier |
|---|---|---|---|---|
| Genome Reference Consortium | 2017 | GRCh38.p12 | https://www.ncbi.nlm.nih.gov/assembly/GCF_000001405.38 | NCBI Assembly, GCA_000001405.27 |
| Max-Planck Institute for Evolutionary Anthropology | 2015 | panpan1.1 | https://www.ncbi.nlm.nih.gov/assembly/GCF_000258655.2 | NCBI Assembly, GCA_000258655.2 |
| University of Washington | 2021 | Mhudiblu_PPA_v2 | https://www.ncbi.nlm.nih.gov/assembly/GCA_013052645.3 | NCBI Assembly, GCA_013052645.3 |
| University of Washington | 2018 | Clint_PTRv2 | https://www.ncbi.nlm.nih.gov/assembly/GCF_002880755.1 | NCBI Assembly, GCA_002880755.3 |
| Wellcome Trust Sanger Institute | 2014 | gorGor4 | https://www.ncbi.nlm.nih.gov/assembly/GCF_000151905.2 | NCBI Assembly, GCA_000151905.3 |
| University of Washington | 2018 | Susie_PABv2 | https://www.ncbi.nlm.nih.gov/assembly/GCF_002880775.1 | NCBI Assembly, GCA_002880775.3 |
| Vervet Genomics Consortium | 2014 | Chlorocebus_sabeus 1.1 | https://www.ncbi.nlm.nih.gov/assembly/GCF_000409795.2 | NCBI Assembly, GCF_000409795.2 |
| Human Genome Sequencing Center | 2017 | Panu_3.0 | https://www.ncbi.nlm.nih.gov/assembly/GCF_000264685.3 | NCBI Assembly, GCF_000264685.3 |
| Baylor College of Medicine | 2015 | Cang.pa_1.0 | https://www.ncbi.nlm.nih.gov/assembly/GCF_000951035.1 | NCBI Assembly, GCF_000951035.1 |
| Laboratory for conservation and utilization of Bio-resource | 2016 | ASM169854v1 | https://www.ncbi.nlm.nih.gov/assembly/GCF_001698545.1 | NCBI Assembly, GCF_001698545.1 |
| Washington University (WashU) | 2013 | Macaca_fascicularis_5.0 | https://www.ncbi.nlm.nih.gov/assembly/GCF_000364345.1 | NCBI Assembly, GCF_000364345.1 |
| Baylor College of Medicine | 2015 | Mnem_1.0 | https://www.ncbi.nlm.nih.gov/assembly/GCF_000956065.1 | NCBI Assembly, GCF_000956065.1 |

*Continued on next page*

*Continued*

| Author(s) | Year | Dataset title | Dataset URL | Database and Identifier |
|---|---|---|---|---|
| Novogene | 2014 | Rrox_v1 | https://www.ncbi.nlm.nih.gov/assembly/GCF_000769185.1 | NCBI Assembly, GCF_000769185.1 |
| The Genome Institute at Washington University School of Medicine | 2019 | Mmul_10 | https://www.ncbi.nlm.nih.gov/assembly/GCF_003339765.1 | NCBI Assembly, GCF_003339765.1 |
| Human Genome Sequencing Center - BCM | 2015 | Caty_1.0 | https://www.ncbi.nlm.nih.gov/assembly/GCF_000955945.1 | NCBI Assembly, GCF_000955945.1 |
| Broad Institute | 2011 | SaiBol1.0 | https://www.ncbi.nlm.nih.gov/assembly/GCF_000235385.1 | NCBI Assembly, GCF_000235385.1 |
| Washington University (WashU) | 2010 | Callithrix jacchus-3.2 | https://www.ncbi.nlm.nih.gov/assembly/GCF_000004665.1 | NCBI Assembly, GCF_000004665.1 |
| Baylor College of Medicine | 2017 | Anan_2.0 | https://www.ncbi.nlm.nih.gov/assembly/GCF_000952055.2 | NCBI Assembly, GCF_000952055.2 |
| McDonnell Genome Institute - Washington University | 2016 | Cebus_imitator-1.0 | https://www.ncbi.nlm.nih.gov/assembly/GCF_001604975.1 | NCBI Assembly, GCF_001604975.1 |
| IBE(CSIC-Universitat Pompeu Fabra) | 2013 | Whole genome sequences for a set of 79 great ape individuals. Genome sequencing | https://www.ncbi.nlm.nih.gov/bioproject?LinkName=biosample_bioproject&from_uid=1920513 | NCBI BioProject, PRJNA189439 |
| Great Ape Genome Project | 2019 | greatape/data | https://eichlerlab.gs.washington.edu/greatape/data/ | greatape, greatape |

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
