## [Editor Report]

In this very interesting manuscript, Baker et al., investigates the molecular evolution in primates of one protein family, the CEACAMs, that are a recurrent target of bacterial surface adhesions at epithelial surfaces. They show that multiple members of this gene family have experienced repeated episodes of positive selection in primates, especially in the N-terminal domains that are associated with protein binding and go on to evaluate the functional consequences of these evolutionary changes. These findings are important to inform our understanding of the co-evolution of interactions between microbes and their mammalian hosts.

---

## [Decision Letter]

**Decision letter after peer review:**

Thank you for submitting your article "Evolution of host-microbe cell adherence by receptor domain shuffling" for consideration by *eLife*. Your article has been reviewed by 3 peer reviewers, and the evaluation has been overseen by us (Stallings and Sawyer) as Reviewing Editor and Senior Editor. The following individuals involved in review of your submission have agreed to reveal their identity: Larry Dishaw (Reviewer #2); Hank Seifert (Reviewer #3).

Essential revisions:

1. The authors should refrain from the arms race analogy and focus on the CEACAM evolution without providing that as the context. The authors conclude the evolutionary pressure is due to escape from bacterial (and possibly other) pathogens but the bacterial species known to bind CEACAMs are host-adapted, opportunistic pathogens (so-called pathobionts) that do not cause significant pathology in hosts and have coevolved with their hosts. Gene conversion is a natural consequence of sequence similarity among paralogous members of a gene family, and it does not seem that special consequences for pathogen evasion need to be invoked. Instead, the presence of frequent gene conversion among members of the CEACAM family seem to indicate that maintaining some degree of sequence similarity among paralogs is beneficial, perhaps due to the role of CEACAM3 as decoy receptor. Moreover, it is uncertain whether the specific bacterium used in this study, Neisseria gonorrhoeae, has been associated with primates at all and how long it's been associated with humans is under debate and suggests this organism is a major driver of CEACAM evolution can be discussed but is not solid. It is more likely that one or more primordial commensal Neisseria species have been present in primates and humanoids. The authors have also ignored the phase variable nature of the Neisseria Opas and the different specificity for CEACAM orthologues.

2. Given that the model is based on a Neisseria species, showing a spirochete form in the model figures is not accurate.

3. In Figure 1: Do the different shades of gray in Figure 1A indicate anything? The legend simply lists gray = all other domains.

4. In Figure 4: The colors of CEACAM 1/3/5/6 are too similar, making Figure 4C hard to interpret. Using more distinctive colors here would be helpful. Just a cladogram could be clearer here, as opposed to having both a tree with and without branch lengths (which is what appears is going on in panel C).

5. Figure 6, panel D appears to be missing.

6. The authors make a point of using GARD to detect recombination breakpoints and accounting for them when they run evolutionary models, e.g. on line 26, page 29. However, it is not exactly clear to me what the authors mean by a "GARD informed phylogeny". Could this be clarified?

7. There have been several other papers on the role of sequence divergence in CEACAM binding, most notably Adrian et al., 2019. While the authors of this paper cite the appropriate references, just a bit more attention to what is novel in this work and what is confirmatory with prior work would be helpful for readers.

8. The statement on line 16, page 21 “gene conversion cannot only serve to maintain CEACAM3’s mimicry function” is well supported. The lack of a CEACAM3 homolog in New World monkeys does not rule out the possibility that CEACAM5 or CEACAM6 functions as a mimic in these species, and the occasional conversion event between non-CEACAM3 proteins certainly could be a byproduct of their sequence similarity (as a consequence of conversion with CEACAM3).

9. The intro would benefit from a bit more detail on the diversity of CEACAMs and mentioning the number of genes in humans (12), and that some are GPI anchored, others have transmembranes, etc. and that this variation influences the diverse physiological functions. Maybe make clear that there appears to be “pathogen-preferred” receptors such as CEACAM1, 3, 5, and 6 in humans, which is not mentioned until the Results section.

10. The authors could mention that some bacterial adhesins have Ig-like domains and this likely influences heterodimer formation, i.e., binding, with the Ig domains of CEACAMs. Various examples exist, Sorge et al., EMBO J, 40:e106103, 2021; also recently reviewed in, Chatterjee et al., FEMS Micro Letters, 368, 2021. The introduction could be a good place to add this, since the case is being made that adhesins tend to interact with CEACAMs.

---

## [Author Response]

Essential revisions:1. The authors should refrain from the arms race analogy and focus on the CEACAM evolution without providing that as the context.

We have modified language in the Introduction in relation to the arms race analogy (lines 37- 42).

The authors conclude the evolutionary pressure is due to escape from bacterial (and possibly other) pathogens but the bacterial species known to bind CEACAMs are host-adapted, opportunistic pathogens (so-called pathobionts) that do not cause significant pathology in hosts and have coevolved with their hosts.

We agree that it is difficult to make specific inferences about the particular microbes that may have driven selection in CEACAMs. We have added language to the Introduction clarifying why we believe that such microbes could drive host adaptation and also that “pathobionts” are not the only microbes that have been found to engage CEACAM proteins (lines 72 – 78).

Gene conversion is a natural consequence of sequence similarity among paralogous members of a gene family, and it does not seem that special consequences for pathogen evasion need to be invoked. Instead, the presence of frequent gene conversion among members of the CEACAM family seem to indicate that maintaining some degree of sequence similarity among paralogs is beneficial, perhaps due to the role of CEACAM3 as decoy receptor.

We now explicitly discuss high sequence identity as a contributing factor for the high frequency of gene conversion observed for some CEACAMs (lines 409 – 412). We have also rewritten a section of the Discussion to include a more explicit example of how exchange of amino acids between epithelial CEACAMs can alter bacterial adhesion (lines 412 – 416).

Moreover, it is uncertain whether the specific bacterium used in this study, Neisseria gonorrhoeae, has been associated with primates at all and how long it’s been associated with humans is under debate and suggests this organism is a major driver of CEACAM evolution can be discussed but is not solid. It is more likely that one or more primordial commensal Neisseria species have been present in primates and humanoids.

We hypothesize that CEACAM evolution has most likely been shaped by multiple microbes over millions of years, and now further elaborate on this point in the Discussion (lines 426 – 429). We also explicitly state that the use of *H. pylori* and *Neisseria* adhesins in this study provides a model for how CEACAM variation can impact diverse types of adhesins, rather than making direct conclusions about the role of these particular microbes in driving host evolution (lines 426 – 431). We have also modified language comparing the evolution of Opa and CEACAM proteins by gene conversion (lines 431 – 434).

The authors have also ignored the phase variable nature of the Neisseria Opas and the different specificity for CEACAM orthologues.

We have added information on the diverse specificities of *Neisseria* Opa proteins to the introduction of our experimental system in the Results section (lines 174 -176). The phase variable nature of *Neisseria* Opa proteins was mentioned in the original draft (now lines 176 – 179).

2. Given that the model is based on a Neisseria species, showing a spirochete form in the model figures is not accurate.

The model was meant to depict a general mechanism by which gene conversion can rapidly alter interactions with bacterial adhesins. It was our intention to depict a *Helicobacter* species in this model, however we agree that not only might the figures appear to depict a spirochete, but also may imply we are arguing this is a mechanism specific to interactions between a limited set of organisms. Consequently, we have changed Figures 1 and 7 to now show an image of a generic rod-shaped bacterium.

3. In Figure 1: Do the different shades of gray in Figure 1A indicate anything? The legend simply lists gray = all other domains.

Different shades of gray indicate different CEACAM domains. To clarify which domains are indicated by different shades we now include this information within the Figure 1 legend.

4. In Figure 4: The colors of CEACAM 1/3/5/6 are too similar, making Figure 4C hard to interpret. Using more distinctive colors here would be helpful. Just a cladogram could be clearer here, as opposed to having both a tree with and without branch lengths (which is what appears is going on in panel C).

In Figure 4 and Figure 4 —figure supplements 2-6 we have changed the colors of CEACAM1/3/5/6 branches on phylogenetic trees to increase contrast and readability. We have also removed the tree with branch lengths from panel C in Figure 4, keeping only the cladogram. We indicate in the figure legend that a cladogram is being depicted in panel C.

5. Figure 6, panel D appears to be missing.

We thank the reviewer for bringing this to our attention. The reference to a panel D in the legend refers to an earlier version of this figure. Text referring to panel D has now been removed from the Figure 6 legend.

6. The authors make a point of using GARD to detect recombination breakpoints and accounting for them when they run evolutionary models, e.g. on line 26, page 29. However, it is not exactly clear to me what the authors mean by a “GARD informed phylogeny”. Could this be clarified?

We have added language clarifying that GARD uses predicted breakpoints to construct phylogenies of gene segments which we then used for subsequent analyses (lines 135 -137 and 639 – 643).

7. There have been several other papers on the role of sequence divergence in CEACAM binding, most notably Adrian et al., 2019. While the authors of this paper cite the appropriate references, just a bit more attention to what is novel in this work and what is confirmatory with prior work would be helpful for readers.

We have modified the final paragraph of the introduction to better define what differentiates our study from previous work on CEACAM evolution (lines 99 – 109).

8. The statement on line 16, page 21 "gene conversion cannot only serve to maintain CEACAM3's mimicry function" is well supported. The lack of a CEACAM3 homolog in New World monkeys does not rule out the possibility that CEACAM5 or CEACAM6 functions as a mimic in these species, and the occasional conversion event between non-CEACAM3 proteins certainly could be a byproduct of their sequence similarity (as a consequence of conversion with CEACAM3).

We agree that we cannot completely rule out that one or more New World monkey CEACAM paralogs serves a function similar to CEACAM 3, and have now stated this in the Discussion section (lines 406 – 407). The comment on sequence similarity facilitating periodic gene conversion is more fully addressed in our response to Essential Revision #1C (manuscript lines 409 – 411)

9. The intro would benefit from a bit more detail on the diversity of CEACAMs and mentioning the number of genes in humans (12), and that some are GPI anchored, others have transmembranes, etc. and that this variation influences the diverse physiological functions. Maybe make clear that there appears to be "pathogen-preferred" receptors such as CEACAM1, 3, 5, and 6 in humans, which is not mentioned until the Results section.

We have added more detail the diversity of CEACAMs and the number of such genes in humans (lines 51 – 54), as well as more information on the membrane anchoring domains and their influence on CEACAM function (lines 60 – 62) to the Introduction. We now also include in the Introduction that CEACAMs 1, 3, 5, and 6 are specifically known to be targeted by bacterial adhesins (lines 65 – 72) and have also better emphasized that these CEACAMs are known to be bacterially targeted in the Discussion (lines 377 – 379 and 416 – 417)

10. The authors could mention that some bacterial adhesins have Ig-like domains and this likely influences heterodimer formation, i.e., binding, with the Ig domains of CEACAMs. Various examples exist, Sorge et al., EMBO J, 40:e106103, 2021; also recently reviewed in, Chatterjee et al., FEMS Micro Letters, 368, 2021. The introduction could be a good place to add this, since the case is being made that adhesins tend to interact with CEACAMs.

As suggested, we have provided additional information in the Introduction noting that certain adhesins use Ig domains to interact with CEACAMs (lines 81 – 84) and CEACAM adhesin interactions could impact native CEACAM dimerization (lines 95 – 98). We thank the reviewers for bringing the Chatterjee *et al.,* review to our attention and have clarified in the introduction that CEACAM proteins are only one of the many host molecules that bacterial adhesins interact with, citing the Chatterjee review where appropriate (lines 48 – 50).